# Trajectory inference from single-cell genomics data with a process time model

**Meichen Fang** [ID][1], **Gennady Gorin** [ID][1,2¤], **Lior Pachter** [ID][1,3]*

**1** Division of Biology and Biological Engineering, California Institute of Technology, Pasadena, California, United States of America, **2** Division of Chemistry and Chemical Engineering, California Institute of Technology, Pasadena, California, United States of America, **3** Department of Computing and Mathematical Sciences, California Institute of Technology, Pasadena, California, United States of America

¤ Current address: Fauna Bio, Emeryville, California, United States of America
* lpachter@caltech.edu

**Data availability statement:** The data sets analyzed in this paper are from previously published research, and are publicly available. The 10x Genomics 10k Human PBMC data can

## Abstract

Single-cell transcriptomics experiments provide gene expression snapshots of heterogeneous cell populations across cell states. These snapshots have been used to infer trajectories and dynamic information even without intensive, time-series data by ordering cells according to gene expression similarity. However, while single-cell snapshots sometimes offer valuable insights into dynamic processes, current methods for ordering cells are limited by descriptive notions of "pseudotime" that lack intrinsic physical meaning. Instead of pseudotime, we propose inference of "process time" via a principled modeling approach to formulating trajectories and inferring latent variables corresponding to timing of cells subject to a biophysical process. Our implementation of this approach, called Chronocell, provides a biophysical formulation of trajectories built on cell state transitions. The Chronocell model is identifiable, making parameter inference meaningful. Furthermore, Chronocell can interpolate between trajectory inference, when cell states lie on a continuum, and clustering, when cells cluster into discrete states. By using a variety of datasets ranging from cluster-like to continuous, we show that Chronocell enables us to assess the suitability of datasets and reveals distinct cellular distributions along process time that are consistent with biological process times. We also compare our parameter estimates of degradation rates to those derived from metabolic labeling datasets, thereby showcasing the biophysical utility of Chronocell. Nevertheless, based on performance characterization on simulations, we find that process time inference can be challenging, highlighting the importance of dataset quality and careful model assessment.

## Author summary

Single-cell RNA sequencing can measure the amounts of RNA in individual cells, and although it is a snapshot experiment, cells that are differentiating can be captured in distinct states allowing for inference of "trajectories" or "velocity". Currently, methods

be downloaded from https://zenodo.org/record/7388133/files/GP_2021_3_raw_data.tar.gz?download=1 with metadata from https://doi.org/10.22002/7fbwg-rx843. The forebrain data can be downloaded from http://pklab.med.harvard.edu/velocyto/hgForebrainGlut/hgForebrainGlut.loom. The erythroid data can be downloaded from https://ndownloader.figshare.com/files/27686871. The RPE1 and Neuron data can be downloaded from the dynamo packages https://github.com/aristoteleo/dynamo-release. All code used to generate the results and figures in the paper is available at https://github.com/pachterlab/FGP_2024.

**Funding:** The author(s) received no specific funding for this work.

**Competing interests:** The authors have declared that no competing interests exist.

that attempt to do so rely heavily on heuristics, with no mechanistic meaning associated with the "pseudotime" they assign to cells. We show that it is possible to infer trajectories under a biophysical model within a principled framework. By developing a trajectory model based on cell state transitions, we demonstrate that it is possible to infer interpretable latent variables, i.e. process time, corresponding to the timing of cells subjected to a biophysical process, as well as transcriptional parameters with biophysical meaning. However, we find this to be a challenging task. By characterizing failure scenarios in simulations and with quantitative assessment on real datasets, we concluded such inference is not always possible, especially when there is insufficient dynamical information embedded in the data. In such cases, our trajectory model allows us to perform model selection to determine if captured cells are better modeled by clusters. Our findings emphasize the importance of thoughtful experimental design and meticulous model assessment for valid trajectory inference.

## Introduction

Single-cell RNA sequencing (scRNA-seq) has provided unprecedented insights into biological dynamical processes in which cells display a continuous spectrum of states that go beyond the confines of discrete cell types [1]. Cells appear to be inherently desynchronized in cellular processes and scRNA-seq can potentially capture cells at different positions over the process even if samples are collected at only one time point. The concept of pseudotime has been developed to describe the position of a cell along the underlying process [2], and trajectory inference (or pseudotemporal ordering) methods aim to solve the inverse problem of inferring the latent pseudotime variable from scRNA-seq data. In light of this this concept, hundreds of methods have been developed [3–13]. However, with a few exceptions that explicitly model gene expression dynamics [11–13], trajectory inference methods mostly treat pseudotime as a descriptive concept relying on more or less arbitrary distance metrics in gene expression space. Specifically, there is no well-defined, agreed-upon meaning underlying the notion of pseudotime, and its interpretation is primarily accomplished through qualitative visuals and low dimensional embeddings.

While a descriptive approach can be powerful in exploratory data analysis, the absence of a well-posed definition for a trajectory renders model interpretation and assessment challenging, even conceptually. Firstly, assessing the credibility of results is hard, as fitting can be performed on any dataset and we have limited metrics and ground truth available to gauge the fit quality. Secondly, the interpretation of the inferred trajectory is not straightforward, and downstream analysis based on pseudotime is employed to understand the underlying gene dynamics. However, this need for following analysis to interpret results gives rise to the problem of circularity (Section 1 in S1 Text), which becomes evident in the context of an inflated false positive rate in the problem of detecting differentially expressed (DE) genes along pseudotime. The problem of circularity is conceptually challenging and can only be effectively remedied under restrictive assumptions [14]. To illustrate these two points, we applied the procedure of trajectory inference and DE analysis on simulations generated from four clusters and were able to "discover" superficially plausible dynamics (S1 Fig). As naive an example as it is, it reflects the fact that we do not have a reliable way to determine the validity of trajectory inference results. Though both are false positives, there is a subtle difference between the falsely inferred trajectory (S1b Fig) and the inflated false positive rate in DE analysis resulting from circularity (S1c Fig): the first one arises when a trajectory model is inferred from

cluster data without proper assessment, while circularity stems from the double use of data for fitting and testing (double dipping) [15]. Conversely, adopting a model-based approach has the potential to mitigate this problem. With a clearly defined model of gene expression along a trajectory, the interpretation of parameters and the characterization of errors becomes more straightforward. First of all, model assessment can be conducted in a more principled manner. We can effectively address the first kind of false positive using conceptually simpler approaches, such as comparing our model to cluster models to identify the correct model. In addition, the specific question of interest like finding DE genes can be incorporated directly into the formulation of the model, rendering ad hoc analysis unnecessary. For example, if we have an probabilistic model of trajectories with transcription kinetics parameters, we can directly select DE genes using inferred parameters, without the need to go through the circular process of fitting trajectories and performing DE analysis to find interesting genes. However, we emphasize that the exact p-values still can not be easily calculated, and circularity persists if we fit trajectories and perform DE test based on the inferred time, which still falls under the issue of double dipping.

Recently, equipped with a kinetic model of RNA dynamics, RNA velocity has emerged as another powerful concept to provide complementary information about dynamic processes [16]. By distinguishing unspliced and spliced mRNA counts as derived from unique molecular identifiers (UMIs) and fitting gene-wise parameters under a on-off model of transcription, it is able to predict the direction of future spliced counts changes. Although a time-dependent gene expression model was explicitly defined in RNA velocity, the time did not have any associated interpretation. Moreover, earlier methods often modeled genes separately with gene-wise times and fit these models after applying a series of ad hoc transformations to count data, which added excessive flexibility and hindered a clear interpretation of the time. As the velocities of different genes had non-comparable scales, they needed to be combined heuristically in a lower-dimensional space to calculate a velocity for a cell [17]. A natural extension is to integrate the cell-wise time of trajectory inference with the mRNA dynamical model of RNA velocity, which a few methods have successfully implemented with different underlying transcription models [18–20]. Moreover, the recent *VeloCycle* developed an RNA velocity model for the cell cycle that models unspliced and spliced counts dynamics directly with harmonic functions [21].

However, despite the implicit pseudotime modeling performed by some of the RNA velocity methods, there remain many challenges in attaching a physical meaning to pseudotime. Do the parameters of the trajectory model have underlying biophysical interpretations? How can we guarantee that our inferences align with our intended objectives? Are the assumptions of trajectory models satisfied to maintain the consistency of our inference? For instance, the application of trajectory inference or RNA velocity methods relies on the assumption of continuous dynamics in the data, which is not examined retrospectively. Though some heuristic scores purport to distinguish between cluster-like data and trajectory-like data [22], there is no principled approach to determine whether the data is sufficiently dynamical and whether a cluster or trajectory model is more appropriate, and the decision of applying trajectory analysis often hinges on prior knowledge and assumptions about the data.

In summary, to attach real meaning to "pseudotime" requires more than just a definition of cell-wise pseudotime. It necessitates a principled approach to statistics to ensure a meaningful inference, which is still lacking in the field of trajectory inference. Meticulous model assessment is required to ensure its relevance to the underlying biological processes and the reliability of results, which includes examining the identifiability of the model, characterizing performance to identify both ideal and failure scenarios, and establishing proper metrics for result falsification. Then pseudotime starts to have a physical meaning, which we suggest

defining as "process time" to underscore its interpretation with respect to a specific cellular process.

The physical interpretation of process time is related to, but not necessarily equivalent to, physical time. Specifically, assuming that all cells share the same dynamic process, we can select a specific point along this process to serve as the starting point for all cells. At the physical sampling time, the process time denotes the relative time to that starting point, indicating how long ago in physical time the cells were at the starting point. Therefore, if the experiments establish a known starting time for when the cells enter the process, the process time should correspond to the relative physical time. On the other hand, if we could follow one cell over time, the process time would evolve in sync with physical time. In reality, where only different cells can be sampled at multiple time points, the distribution of process time should ideally evolve in parallel with physical time, provided enough cells are sampled from the same population.

Here, we build such a model and infer "process time" in a principled way with Chronocell. To strike a balance between expressiveness and identifiability, we proposed a trajectory model built on cell states [17]. On the one hand, we incorporated different cell states so that our trajectory model is expressive enough to capture the observation that cells are generally assumed to transition through various states during development. On the other hand, we assume a constant transcription rate for each state to keep the model as simple as possible. By introducing simplifying assumptions in transcription and sequencing model, we ensure model identifiability. We consider the influence of technical aspects and directly incorporate them into the distribution of counts, which eliminates the necessity for unjustified heuristic preprocessing steps that lead to unclear interpretation and biased results even in the large data and no noise limit [17]. We undertook simulations to characterize estimation accuracy in the different parameter regimes to identify ideal and failure scenarios. Using simulations for which ground truth is known allowed us to characterize how large uncertainty and inconsistent parameter values serve as good indicators of potential unreliability in failure scenarios, which can be assessed even when ground truth is unavailable. Finally, we applied Chronocell to biological datasets. We assessed its appropriateness on different datasets and identified unsuitable ones. For suitable datasets, Chronocell revealed distinct cellular distributions over process time and yielded mRNA degradation rate estimates congruous with those obtained from mRNA metabolic labeling.

## Methods

We begin by describing our trajectory model, followed by a description of the inference procedure. Next, we explain the analysis pipeline, including our gene and model selection procedure. Then, we elucidate the simulation setup. Finally, we detail the preprocessing of real datasets.

Throughout the Methods we denote data by $\mathbf{X}$, and note that $\mathbf{X}$ corresponds to a cell by gene by species array. We use $i$ to index cells, $j$ to index genes, and $c$ to index species. Parameters are denoted by $\theta$, and latent variables by $z$. $p(\cdot)$ means a probability distribution.

### Model

We used a simple and interpretable latent variable model for the probability distribution of the counts, with explicit biophysical meaning associated to each latent variable. We assumed cells are asynchronous, and we therefore introduced two latent variables that corresponded to the lineage and time of the cell.

Therefore, in our trajectory model, the gene expression data of each cell $\mathbf{x}$ was described as a function of latent variables $\mathbf{z} = (l, t)$ which specify the lineage $l$ and time $t$ of the cell. By considering a particular parametric class of gene expression dynamics that specify the distribution $p(\mathbf{y}|\mathbf{z}, \theta)$ of in vivo counts $\mathbf{y}$, and a sequencing noise model $p(x|y)$, we mapped latent variable $z$ into data space and arrived at an explicit formulation of data distribution that could be trained using the expectation-maximization (EM) algorithm. In other words, we assumed the counts of each cell $\mathbf{x}$ had the following density,

$$p(\mathbf{x}|\theta) = \int_{\mathbf{z}} p(\mathbf{x}|\mathbf{z}, \theta)p(\mathbf{z})d\mathbf{z} = \int_{\mathbf{z}} p(\mathbf{x}|\mathbf{y})p(\mathbf{y}|\mathbf{z}, \theta)p(\mathbf{z})d\mathbf{z},$$

where $p(z)$ is the distribution of latent variables.

In the following, we specify $p(y|z, \theta)$, $p(x|y)$ and $p(z)$, which are based on the transcription model, measurement model and sampling measure.

**Transcription model**   Since current transcriptomic data can be used to measure the numbers of both nascent and mature transcripts, we considered the production, processing, and degradation of individual RNA molecules in our transcription model, as in previous RNA velocity literature [16].

The distribution of RNA counts in the reaction system (Eq 2) is known [23]. We assumed that the initial distribution of $U$ and $S$ was a product Poisson distributions, which implied that the mean parameter vector $\boldsymbol{\lambda} = (\lambda_u, \lambda_s)$ of $U$ and $S$ evolved according to Eq (3).

The functional form of $A_l(t)$ reflects our assumptions of gene expression during development. However, explicitly modeling transcription rates as functions of gene expressions is hard and tends to overfit. More fundamentally, we have some prior physical intuition; this function is actually $A_l(y, \mathbf{u}, t)$, where $\mathbf{u}$ is some high-dimensional vector of regulator concentrations. These data are not possible to collect using currently available technologies, although this constraint may change in the coming years. In principle, we can write down these equations, and even simulate them (using dyngen [24] or the stochastic simulation algorithm), but we strived to start with an analytically tractable model, particularly to recapitulate and formalize the increasingly popular RNA velocity framework. Therefore, we used a phenomenological model and didn't consider gene interactions. Rather, the effect of transcriptional regulation on each gene during development was summarized into synchronized state switching and each state had its transcription rate. This formalized transient cell types.

To be able to account for multiple lineages, we assumed cell states $\mathcal{S} = 0, 1, ..., S$ formed a directed graph, with each lineage represented as a path of length $K$ on this graph.. The trajectory structure described the graph, and recorded the states $s(l, k) \in \mathcal{S}, l = 1, ..., L, k = 1, ..., K$ of the $L$ lineages during the $K$ stages. We treated the trajectory structure as known since it can typically be obtained from our previous knowledge of the data. Specifically, for lineage $l$, we assumed $A_l(t)$ is a piecewise constant function: $A_l(t) = \alpha_s(l, k)$, where the cellular state index s is determined by the lineage $l$, and the time interval $k$ to which $t$ belongs, i.e., $t \in (\tau_{k-1}, \tau_k]$. The state index $s = s(l, k)$ was determined by the trajectory structure. For example, given the trajectory structure in Fig 1, $s(l = 1, k = 0) = s(l = 2, k = 0) = 0$, $s(l = 1, k = 1) = s(l = 2, k = 1) = 1$, $s(l = 2, k = 2) = 2$ and $s(l = 2, k = 2) = 3$.

## Chronocell

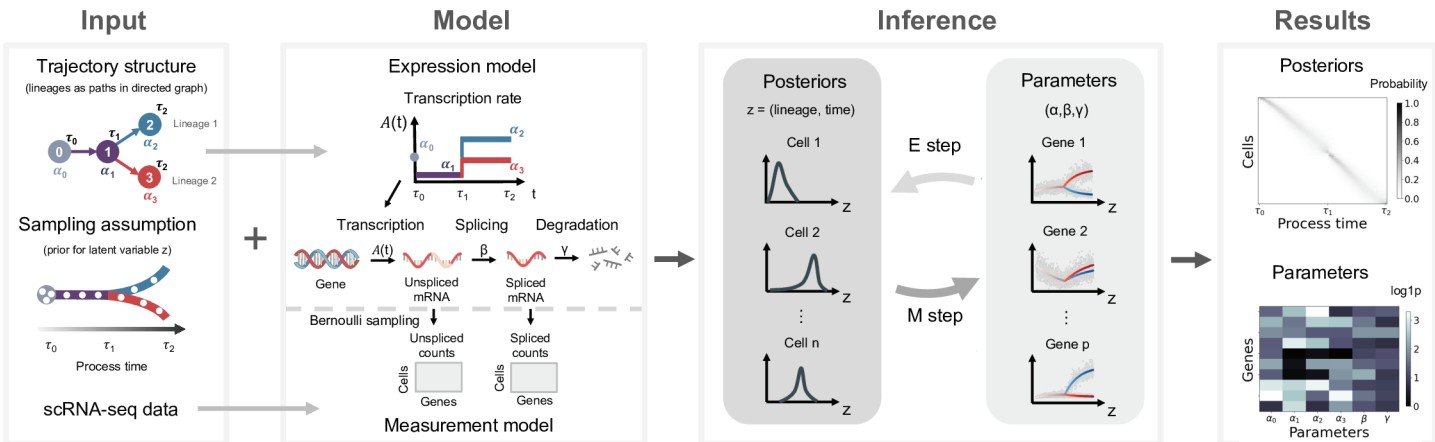

**Fig 1. Chronocell overview**. The **input** of Chronocell comprises three components: 1) the *trajectory structure*, which outlines the states each lineage traverses as paths on a directed graph. 2) The *sampling assumption*, which defines the prior distribution of latent variables, namely lineages and process time, with a default uniform distribution over both; and 3) the *scRNA-seq data*, consisting of unspliced and spliced count matrices. The Chronocell **model** consists of a *expression model* with piecewise-constant transcription rates, and a Bernoulli *measurement model*. Each state $s$ is associated with a transcription rate $\alpha_s$ for each gene, as well as an exit time $\tau_k$ denoting the switching time to the next state, where k is the index for the time segment. The EM algorithm is used for **inference**, with each iteration alternating between E-steps and M-steps. The **results** of Chronocell primarily include the estimated parameters and posterior distributions over latent variables for each cell.

Then, the parametric solution of $\lambda$ was given by

$$q = \arg\min_{k}\{k|\tau_k \geq t\},$$

$$\lambda_u(t) = \sum_{k=1}^{q-1} \frac{\alpha_{s(l,k)}}{\beta}\left(e^{-\beta(t-\tau_k)} - e^{-\beta(t-\tau_{k-1})}\right) + \frac{\alpha_{s(l,q)}}{\beta}\left(1 - e^{-\beta(t-\tau_{q-1})}\right) + \lambda_u(0)e^{-\beta t},$$

$$\lambda_s(t) = \sum_{k=1}^{q-1} \frac{\beta\alpha_{s(l,k)}}{\gamma(\beta-\gamma)}\left(e^{-\gamma(t-\tau_k)} - e^{-\gamma(t-\tau_{k-1})}\right) + \frac{\beta\alpha_{s(l,q)}}{\gamma(\beta-\gamma)}\left(1 - e^{-\gamma(t-\tau_{q-1})}\right)$$

$$+ \left(\lambda_s(0) + \frac{\beta}{\beta-\gamma}\lambda_u(0)\right)e^{-\gamma t} - \frac{\beta}{\beta-\gamma}\lambda_u(t).$$

Assuming cells are at steady states at time 0, we have $\lambda_u(0) = \alpha_{s_{l,0}}$ and $\lambda_s(0) = \frac{\beta}{\gamma}\alpha_{s_{l,0}}$. Therefore, for each gene, the parameters are $\theta = (\alpha, \beta, \gamma, \tau)$.

**Measurement model** Next, we needed to have a measurement model that connected counts number **y** in cells to the observed counts **x** in a single-cell RNA-seq experiment.

We assumed each molecule of mRNA produces *Bernoulli(q)* number of captured RNA molecules, which is also an good approximation of a Poisson model with low mean [25,26]. The capture rate $q$ for each molecule can potentially depend on factors such as cell read depth, gene-specific, and species-specific biases in mRNA capture methods. However, in our model, we assume the capture rate only varies with cell read depth (or cell size), i.e., $q = r_i$, where $r_i$

represents the read depth (cell size) of cell $i$. With $i$ as the cell index, $j$ as the gene index, and $c$ as the species index, we have:

$$x_{ijc} \sim binomial(y_{ijc}, r_i), \quad y_{ijc} \sim Poisson(\lambda_{jc}),$$

where $\lambda_{jc}$ is the ODE solution for species c of gene j.

This is equivalent to multiplying the mean of Poisson distributions by a constant $r_i$. Since usually only the relative value of read depth $r_i$ can be available, we absorbed the mean of $r_i$ into $\alpha$ and infer their product directly:

$$x_{ijc} \sim Poisson(r_i \lambda_{jc}).$$

**Sampling distribution** Now that we have defined the $p(x|z, \theta)$, the only remaining thing to complete $p(x|\theta)$ is the sampling distribution $p(z)$, which describes the prior distribution of latent variables. Given the formula of $p(y|z, \theta)$, it is easy to see that the model is not identifiable if $p(z)$ is not fixed, because we can change $p(z)$ together with $\beta$ and $\gamma$ easily without changing $p(x)$, for example, by scaling $\beta$ and $\gamma$ and transform $p(z)$ accordingly. Therefore, we assumed $t \in [0, 1]$ and fixed a uniform prior for $t$ on $(0, 1)$. With this given prior distribution, the model was identifiable, because the parameterized form of the means of Poisson distributions is identifiable and Poisson distribution is identifiable [27]. If one has information about real time, one can adjust the range and prior of $t$ to be have more physical meaning. For example, for the cell cycle dataset, if one knows that the whole cycle takes 24 hours, then one can either set the range of $t$ to be $[0, 24]$, or scale the results by dividing both $\beta$ and $\gamma$ by 24, while keeping the other parameters unchanged.

**Connection to cluster models** In the fast dynamic limit, as there are few cells out of steady states, transitions (edge) disappear and only states (nodes) remain, and both the weight of lineages and the length of time interval (or the weight at t=0 for state 0) determine the mixture weights of clusters. Specifically, for the state 0, its weight equals the weight at t=0, while for the state 2, its weight equals the product of the length of the second time interval $(\tau_1, \tau_2]$ and the weight of the second lineages. Thus, the Poisson mixtures model strictly belongs to the degenerate cases of our trajectory model, which connects Chronocell to biophysical cluster models like meK-Means [28], barring differences in noise models and hard/soft assignment. This does not mean that our trajectory model should be fit on cluster data, because the resulted process time is no longer meaningful. Instead, these connections allowed us to better compare the models, and determine which model was more appropriate (Section Gene selection).

## Inference

### Chronocell overview

**Input.** The input of Chronocell are: 1) trajectory structure; 2) sampling assumption; 3) scRNA-seq count matrix. Trajectory structure is provided to Chronocell as a 2D array, with each lineage (path) represented as a row. Along with the structure, an initial guess of switching time is also needed as a starting point in the fitting. The sampling assumption refers to the prior distribution of the latent variables (process time and lineages) for each cell. This is represented as a 3D array with shape (n, L, M), where n is the number of cells, L is the number of lineages, and M is the number of time grids.

**Model.** Building upon the common transcription model, we have two classes of models based on the assumption of global switch time: (1) the synchronized model, which assumes a

completely synchronized switch in transcription rates across all genes; and (2) the desynchronized model, where each gene has its own switching time. The desynchronized model is more challenging to fit from scratch, so we recommend using a warm start based on the results of the synchronized model.

**Inference.** We use the expectation–maximization algorithm to fit the trajectory model on the scRNA-seq count matrix. See Section Maximum likelihood estimates of parameters by EM algorithm.

**Output.** The primary output of Chronocell consists of the parameters and posterior distribution for each cell. Other relevant information such as the Akaike Information Criterion (AIC) and the Fisher information matrix can also be calculated.

**Maximum likelihood estimates of parameters by EM algorithm** We use the expectation–maximization algorithm to estimate model parameters $\hat{\theta}$. For simplicity, we discretize the latent variable $t$ with finite regular grid points over the interval: $t = t_1, ...., t_M$, which basically approximates a continuous measure with a discrete measure. Then, the latent variable z=(l,t) describing lineage and time is discrete, and we write $\sum_{z_i}$ to denote the summation over all L lineages and M time grid points, i.e., $\sum_z = \sum_{l=1}^{L} \sum_{m=1}^{M}$. With this, the objective function becomes

$$\hat{\theta} = \arg\max_{\theta} \log p(x|\theta),$$

$$\log p(x|\theta) = \sum_{i=1}^{n} \log = \sum_{i=1}^{n} \log \int_{z_i} p(x_i, z_i|\theta) dz_i \approx \sum_{i=1}^{n} \log \sum_{z_i} p(x_i, z_i|\theta).$$

where $i$ is the cell index and $p(x_i, z_i|\theta)$ denotes the probability of observing $x_i$ with the latent variable being $z_i$ for cell $i$.

As log function is concave, we can use Jensen's inequality:

$$\sum_{i=1}^{n} \log \sum_{z_i} p(x_i, z_i|\theta) = \sum_{i=1}^{n} \log \sum_{z_i} p(z_i|x_i, \theta) \frac{p(x_i, z_i|\theta)}{p(z_i|x_i, \theta)},$$

$$\geq \sum_{i=1}^{n} \sum_{z_i} p_i(z_i|x_i, \theta) \log \frac{p(x_i, z_i|\theta)}{p_i(z_i|x_i, \theta)}.$$

Since $\frac{p(x_i, z_i|\theta)}{p(z_i|x_i, \theta)} = p(x_i|\theta)$ is a constant for all $z_i$, the equality holds and

$$\sum_{i=1}^{n} \log \sum_{z_i} p(x_i, z_i|\theta) = \sum_{i=1}^{n} \sum_{z_i} p_i(z_i|x_i, \theta) \log \frac{p(x_i|z_i, \theta)p(z_i)}{p_i(z_i|x_i, \theta)}.$$

Our trajectory model makes it possible to write out $p(x_i|z_i, \theta)$ explicitly:

$$\log p(x_i|z_i, \theta) = \sum_{j=1}^{p} \log p(x_{ij}|z_i, \theta_j),$$

$$\log p(x_{ij}|z_i, \theta_j) = \sum_{c=1}^{2} \left[ \left( x_{ijc} \log \left( r_i \lambda_c(l, t_m, \theta_j) \right) - r_i \lambda_c(l, t_m, \theta_j) - \log(x_{ijc}!) \right) \right].$$

Therefore, we could use the expectation–maximization algorithm efficiently. Specifically, in the E-step of EM algorithm, we calculated the posterior distribution $p_i(z_i|x_i,\theta)$ based on $p(x_i|z_i,\theta)$,

$$
\begin{aligned}
p(z_i|x_i,\theta) &= \frac{p(z_i)p(x_i|z_i,\theta)}{\sum_{z_i} p(z_i)p(x_i|z_i,\theta)} \\
&= \frac{p(z_i)\exp\left(\log p(x_i|z_i,\theta)\right)}{\sum_{z_i} p(z_i)\exp\left(\log p(x_i|z_i,\theta)\right)}.
\end{aligned}
$$

In the M-step, based on fixed $p_i(z_i|x_i,\theta)$ and analytical form of $p(x_i|z_i,\theta)$, we optimized $\theta_j$ for each gene $j$ separately by maximizing

$$
F_j = \sum_{i=1}^{n}\sum_{z_i} p_i(z_i|x_i,\theta)\log p(x_{ij}|z_i,\theta_j),
$$

Both the value and the gradient of $F_j$ can be written out analytically and computed efficiently (Section 2 in S1 Text). Therefore, we could use off-the-shelf quasi-Newton methods for optimizing $F_j$ with respect to $\theta_j$, e.g., 'L-BFGS-B' method in minimize function provided by Scipy [29,30].

In each step of EM algorithm, we alternated between the expectation and maximization steps. The implementation is based on defining the function that calculates $p(x_{ij}|z_i,\theta)$, so it would be easy to modify $p(x_{ij}|z_i,\theta)$ for different models in the future.

A warm start incorporating prior knowledge about data can help algorithm converge to the optimal $\theta^*$ quickly. If neither initial parameters nor $p(z|x)$ is given, we use random initializations. Multiple runs with different starting points are used to avoid local minima. By default, we tried 100 different random initializations, and run 100 steps for each initialization both for random initialization and warm start.

**Read depth estimation**  If we refer to all cell-wise variances as extrinsic noise, then it includes both sequencing noise and a part of biological sotchasticity. This extrinsic noise is shared by covariance between genes and variance of genes. Therefore, we can estimate such extrinsic noise using covariance to leave out the intrinsic noise of genes that are more biologically relevant.

Consider two species $a$ and $b$. Suppose that *in vivo* their counts numbers are $Y_a$ and $Y_b$ with $Y_a \sim \mu_a$ and $Y_b \sim \mu_b$ where $\mu_a$ and $\mu_b$ are arbitrary distributions. Suppose additionally that during sequencing, the read depth (cell size) $r$ is a random variable on $[0,1]$, and $r$ is independent of $Y_a$ and $Y_b$. Denote the counts we ultimately get by $X_a$ and $X_b$, where $X_a$ and $X_b$ are independent given $r$, $Y_a$, and $Y_b$, and $\mathrm{E}\left[X_a|r,Y_a\right] = r\,\mathrm{E}\left[Y_a\right]$ and $\mathrm{E}\left[X_b|r,Y_b\right] = r\,\mathrm{E}\left[Y_b\right]$.

$$
\begin{aligned}
\mathrm{Cov}\left(X_a,X_b\right) &= \mathrm{E}\left[\left(X_a - \mathrm{E}\left[X\right]_a\right)\left(X_b - \mathrm{E}\left[X\right]_b\right)\right] \\
&= \mathrm{E}\big[\left(X_a - \mathrm{E}\left[r\right]Y_a + \mathrm{E}\left[r\right]Y_a - \mathrm{E}\left[r\right]\mathrm{E}\left[Y_a\right]\right) \\
&\qquad \left(X_b - \mathrm{E}\left[r\right]Y_b + \mathrm{E}\left[r\right]Y_b - \mathrm{E}\left[r\right]\mathrm{E}\left[Y_b\right]\right)\big] \\
&= \mathrm{E}\left[\left(X_a - \mathrm{E}\left[r\right]Y_a\right)\left(X_b - \mathrm{E}\left[r\right]Y_b\right)\right] \\
&\qquad + \mathrm{E}\left[\left(\mathrm{E}\left[r\right]Y_a - \mathrm{E}\left[r\right]\mathrm{E}\left[Y_a\right]\right)\left(\mathrm{E}\left[r\right]Y_b - \mathrm{E}\left[r\right]\mathrm{E}\left[Y_b\right]\right)\right],
\end{aligned}
$$

$$E\left[(X_a - E[r]Y_a)(X_b - E[r]Y_b)\right] = E[(X_a - rY_a + rY_a - E[r]Y_a)$$
$$(X_b - rY_b + rY_b - E[r]Y_b)]$$
$$= E\left[(rY_a - E[r]Y_a)(rY_b - E[r]Y_b)\right]$$
$$= E[Y_aY_b]\,\mathrm{Var}[r],$$

$$E\left[(E[r]Y_a - E[r]E[Y_a])(E[r]Y_b - E[r]E[Y_b])\right] = \mathrm{Cov}(Y_a, Y_b)\,E[r]^2.$$

Therefore,

$$\mathrm{Cov}(X_a, X_b) = E[Y_aY_b]\,\mathrm{Var}[r] + \mathrm{Cov}(Y_a, Y_b)\,E[r]^2$$
$$= E[Y_a]E[Y_b]\,\mathrm{Var}[r] + \mathrm{Cov}(Y_a, Y_b)\,\mathrm{Var}[r] + \mathrm{Cov}(Y_a, Y_b)\,E[r]^2,$$

and

$$\frac{\mathrm{Cov}(X_a, X_b)}{E[X_a]E[X_b]} = \frac{\mathrm{Var}[r]}{E[r]^2} + \frac{\mathrm{Cov}(Y_a, Y_b)}{E[Y_a]E[Y_b]}\left(1 + \frac{\mathrm{Var}[r]}{E[r]^2}\right).$$

If we assume genes are independent on average, then we can estimate the $\mathrm{CV}^2$ of read depth $\left(\frac{\mathrm{Var}[r]}{E[r]^2}\right)$ by $\frac{\mathrm{Cov}(X_a, X_b)}{E[X_a]E[X_b]}$. We can summarize the extrinsic noise by defining $\xi := E\left[\frac{\mathrm{Cov}(X_a, X_b)}{E[X]_a E[X]_b}\right]_{a,b}$, where the expectation is taken with respect to all possible gene pairs (a,b), and we can estimate $\xi$ by average normalized covariance across genes.

On the other hand, the variance of one gene is

$$\mathrm{Var}[X] = E\left[(X - E[X])^2\right]$$
$$= E\left[(X - rY + rY - E[Y]r + E[Y]r - E[Y]E[r])^2\right]$$
$$= E\left[(X - Yr)^2\right] + E\left[(Yr - E[Y]r)^2\right] + E\left[(E[Y]r - E[Y]E[r])^2\right]$$
$$= E\left[\mathrm{Var}[X|rY]\right] + \mathrm{Var}[r]\,\mathrm{Var}[Y] + E[r]^2\,\mathrm{Var}[Y] + \mathrm{Var}[r]\,E[Y]^2.$$

Now apply a second order approximation and suppose $\mathrm{Var}[X|r, Y] = \eta_1 rY + \eta_2 r^2 Y$ and $\mathrm{Var}[Y] = \phi_1 E[Y] + \phi_2 E[Y]^2$ sequentially.

Then,

$$\mathrm{Var}[X] = E[\mathrm{Var}[X|r, Y]] + \mathrm{Var}[r]\,\mathrm{Var}[Y] + E[r]^2\,\mathrm{Var}[Y] + \mathrm{Var}[r]\,E[Y]^2$$
$$= \eta_1 E[r]E[Y] + \eta_2 E[Y]E[r^2] + \mathrm{Var}[r]\,\mathrm{Var}[Y] + E[r]^2\,\mathrm{Var}[Y] + \mathrm{Var}[r]\,E[Y]^2$$
$$= [\eta_1 + \eta_2(1 + \xi)E[r]]E[X] + \xi E[X]^2 + (1 + \xi)E[r]^2\,\mathrm{Var}[Y].$$

and

$$\mathrm{Var}[X] = [\eta_1 + \eta_2(1 + \xi)E[r]]E[X] + \xi E[X]^2 + (1 + \xi)E[r]^2\,\mathrm{Var}[Y]$$
$$= [\eta_1 + \eta_2(1 + \xi)E[r]]E[X] + \xi E[X]^2 + (1 + \xi)E[r]^2\left(\phi_1 E[Y] + \phi_2 E[Y]^2\right)$$
$$= [\eta_1 + (\eta_2 + \phi_1)(1 + \xi)E[r]]E[X] + \xi E[X]^2 + (1 + \xi)\phi_2 E[X]^2$$
$$= (\eta_1 + (\eta_2 + \phi_1)(\xi + 1)E[r])E[X] + ((\xi + 1)\phi_2 + \xi)E[X]^2.$$

It's easy to see that if $\phi_2 = 0$, i.e., gene intrinsic variance does not have quadratic terms, the coefficient of quadratic term of variance is also $\xi$. Therefore, if the quadratic term of the variance of a gene is closed to $\xi$, we know it is more Poissonian. In other words, it does not differentially expressed and could be used to estimate read depth.

Therefore, our procedure of read depth estimation is: first, estimate read depth $CV^2$ and $\xi$; then, select genes whose quadratic term of the variance is close to $\xi$; finally, sum them up and normalize by the mean to calculate relative read depth. A similar, but not identical procedure of read depth estimation, was used in [31].

**Poisson mixtures model**  For fitting clusters, we used Poisson mixture models. Suppose there are S mixtures with transcription rates $\alpha_{js}, s = 1, ..., S$ for each gene $j$, and each mixture is at steady state. Thus, the parameters of one gene are $a_j = \frac{\alpha}{\beta}$ and $\rho = \frac{\beta_j}{\gamma_j}$, since only the ratio is identifiable.

$$p(x_i|\theta) = \sum_s p(x_i|s, a_{js}, \rho)$$
$$= \sum_s \Pi_j P_{\text{Poiss}}(X = x_{ij1}; \lambda = r_i a_s) P_{\text{Poiss}}(X = x_{ij2}; \lambda = r_i a_{js}\rho).$$

Similarly to the Gaussian mixture model, we could use the EM algorithm to infer parameters (including mixture weights) and posteriors.

## Analysis

**Fisher information**  The Fisher information matrix (FIM) is defined to be

$$\mathcal{I}(\theta_0) \equiv \mathrm{E}\left[\left[\frac{\partial}{\partial\theta}\log(p(x|\theta))\right]\left[\frac{\partial}{\partial\theta}\log(p(x|\theta))\right]^{\mathsf{T}}\right]_{|\theta=\theta_0}.$$

Since

$$\frac{\partial}{\partial\theta}\log(p(x|\theta)) = \frac{\partial}{\partial\theta}\log\left(\int p(x,z|\theta)dz\right)$$
$$\approx \frac{\partial}{\partial\theta}\log\left(\sum_z p(x,z|\theta)\right)$$
$$= \frac{1}{\sum_z p(x,z|\theta)}\sum_z\left(\frac{\partial p(x,z|\theta)}{\partial\theta}\right)$$
$$= \frac{1}{\sum_z p(x,z|\theta)}\sum_z\left(p(z)\frac{\partial e^{\log p(x|z,\theta)}}{\partial\theta}\right)$$
$$= \frac{1}{\sum_z p(x,z|\theta)}\sum_z\left(p(z)e^{\log p(x|z,\theta)}\frac{\partial\log p(x|z,\theta)}{\partial\theta}\right)$$
$$= \sum_z\left(\frac{p(x,z|\theta)}{\sum_z p(x,z|\theta)}\frac{\partial\log p(x|z,\theta)}{\partial\theta}\right)$$
$$= \sum_z\left(p(z|x,\theta)\frac{\partial\log p(x|z,\theta)}{\partial\theta}\right),$$

we could calculate FIM numerically with the explicit form of the derivative and the posterior distribution.

**Gene selection**   We did not use absolute likelihoods to select dynamical genes, because different genes have different scales and are not directly comparable. Instead, we noticed that the cluster model serves as a natural reference point for comparison, as genes with no dynamics can be fit equally well, if not better, by clusters. Therefore, we decided to use the relative likelihood as a criterion for gene selection.

For trajectory model,

$$
\sum_{i=1}^{n} \log \sum_{z_i} p(x_i, z_i | \theta) = \sum_{i=1}^{n} \sum_{z_i} p_i(z_i | x_i, \theta) \log \frac{p(x_i | z_i, \theta) p(z_i)}{p_i(z_i | x_i, \theta)}
$$

$$
= \sum_{i=1}^{n} \sum_{z_i} p_i(z_i | x_i, \theta) \log \frac{p(z_i)}{p_i(z_i | x_i, \theta)}
$$

$$
+ \sum_{i=1}^{n} \sum_{z_i} p_i(z_i | x_i, \theta) \log p(x_i | z_i, \theta)
$$

$$
= \sum_{i=1}^{n} \sum_{z_i} p_i(z_i | x_i, \theta) \log \frac{p(z_i)}{p_i(z_i | x_i, \theta)}
$$

$$
+ \sum_{i=1}^{n} \sum_{j=1}^{p} \sum_{z_i} p_i(z_i | x_i, \theta) \log p(x_{ij} | z_i, \theta_j).
$$

Similarly for clusters model,

$$
\sum_{i=1}^{n} \log \sum_{s=1}^{S} p(x_i, s | \theta) = \sum_{i=1}^{n} \sum_{s} p_i(s | x_i, \theta) \log \frac{p(s)}{p_i(s | x_i, \theta)}
$$

$$
+ \sum_{i=1}^{n} \sum_{j=1}^{p} \sum_{s} p_i(s | x_i, \theta) \log p(x_{ij} | s, \theta_j).
$$

We defined the gene-wise likelihood of gene j for trajectory model to be

$$
\frac{1}{p} \sum_{i=1}^{n} \sum_{z_i} p_i(z_i | x_i, \theta) \log \frac{p(z_i)}{p_i(z_i | x_i, \theta)} + \sum_{i=1}^{n} \sum_{z_i} p_i(z_i | x_i, \theta) \log p(x_{ij} | z_i, \theta_j),
$$

and for clusters,

$$
\frac{1}{p} \sum_{i=1}^{n} \sum_{s} p_i(s | x_i, \theta) \log \frac{p(s)}{p_i(s | x_i, \theta)} + \sum_{i=1}^{n} \sum_{s} p_i(s | x_i, \theta) \log p(x_{ij} | s, \theta_j).
$$

The first term is meant to represent the difference in likelihood caused by different flexibility of latent variables of two models.

For gene selection, we compared the gene likelihood of two models and selected genes whose gene likelihood of trajectory model was higher than that of the cluster model.

**Uncertainty assessment**   We used uncertainty/instability to falsify results. First, we evaluated the variation of different random initializations to assess the uncertainty of the inference. This was done by performing multiple (usually 100) random initializations. Specifically, we evaluated whether the process time estimation of initializations with high ELBO scores concentrated around the correct direction. To quantify this, we classified the output of each random initialization to be correct if the process time estimates had a correlation higher than 0.8

with the reference, and we used different ELBO score thresholds to compute the precision-recall curve, which were then summarized by average precision (AP). An ideal case with high ELBO scores concentrated around correlation one led to an AP close to one while multiple comparable maxima lead to low AP.

Another approach to assess uncertainty is through bootstrap resampling. The same inference procedure is applied to the resampled data, and the variation in the resultant process time serves as an indicator of instability. Specifically, we calculate the correlation of process time between the original and the resampled data and interpret instability as large variance of the correlation.

## Simulations

We randomly generated parameters for 200 genes and sampled 2000 cells by default unless otherwise specified. Cells were sampled uniformly over process time and lineages. We then fit the model with the correct trajectory structure and synchronized model if not stated otherwise.

For the simulation parameters, we assumed that the transcription parameters $(\alpha, \beta, \gamma)$ followed the log-normal distribution $lognormal(\mu, \sigma)$, where $\mu$ is the mean and $\sigma$ is the standard deviation of the variable's natural logarithm. We attempted to derive realistic parameters for the distributions from the literature. Hence, we assumed $\beta \sim lognormal(2, 0.5)$, $\gamma \sim lognormal(0.5, 0.5)$ so that unspliced to spliced ratio was around 0.2 and their distributions resembled those determined from metabolic labeling datasets [32]. The $\alpha$ was assumed to follow $lognormal(2, 1)$, so that the mean of spliced counts was similar to that in scRNA-seq data. We assumed the read depth follows Beta distribution $r \sim Beta(\mu = \frac{1}{4}, v = \frac{1}{64})$, where $\mu$ was the mean and $v$ the variance. For simulations with Gamma noise, we multiplied the mean of Poisson distribution $\lambda$ by a random variable following Gamma distribution with mean 1 and variance 0.5.

For simulations under the desynchronized model, gene-wise $\tau_k$ was sampled from a uniform distribution on $\left[ T_k - \frac{\Delta\tau}{2}, T_k + \frac{\Delta\tau}{2} \right]$, where $T_k$ corresponds to the global $\tau_k$ in the synchronized model, and $\Delta\tau$ is the smallest interval length ensuring that $\tau_k$ retains its order.

To vary the sampling distributions, we generated a Gaussian distribution with a random mean and standard deviation 0.05. We then sampled both from the Gaussian and the uniform distribution, and blended them together in different proportions. The percentages of time sampled from a Gaussian ranged from 0 to 1, increasing by 0.1 increments.

To characterize the time errors, we calculate root mean squared error (RMSE) for the posterior mean of process time in comparison to the true time. To calculate the errors of parameters, we divide the absolute error of $\alpha$ by the square root of true values and $\beta, \gamma$ by true values, so that errors of different genes are more comparable. We refer to these as normalized errors throughout the text.

## Real datasets preprocessing

To estimate the squared coefficient of variation of the read depth $\xi := E\left[ \frac{Cov(X_a, X_b)}{E[X]E[Y]} \right]_{a,b}$, we calculated the covariance matrix of all genes with nonzero means, which was divided by the mean squared and averaged across gene pairs to calculate the mean normalized covariance as an estimate of $\xi$. We then selected Poissonian genes whose variances are close to baseline variance with reasonably large mean ($var < 1.2(\mu + \xi\mu^2), \mu > 0.01$). However, as some genes can be co-regulated and, therefore, correlated, we calculated the mean normalized covariance of Poissonian genes and repeated selecting new Poissonian genes until the mean normalized

covariance no longer changed. This typically occurred after two iterations. Finally, we normalized the sum of counts of the selected Poissonian genes by their mean to obtain the relative read depth estimates, which were then used as fixed parameters during the fitting process.

Based on simulations of factors of informativeness of the data, including numbers of cells and genes, the average mean of $\alpha$ parameters as well as $\beta$ and $\gamma$ ratio, we determined the procedure of filtering genes for fitting: we applied count mean thresholds (0.02 for unspliced and 0.1 for spliced), filtered out genes with a small unspliced to spliced ratio $\left(\frac{U}{S} < e^{-4}\right)$, and finally, selected genes with a variance larger than 1.2 times the baseline variance, i.e., $var > 1.2(\mu + \xi\mu^2)$, where $\mu$ is the mean, $var$ is the variance, and $\xi$ is the read depth $\mathrm{CV}^2$. For cell cycle data, we also constrained genes to the Gene Ontology term "cell_cycle" (GO:0007049). We occasionally adjusted the variance threshold to 1.5 in order to end up with 50–200 genes. Then trajectory and Poisson mixture models were fit on those genes.

### Use of other methods

We have also used dyngen to generate simulation data [24]. We use a bifurcation backbone for generation of 10000 cells and 200 genes following its vignette.

For Monocle 3 and diffusion pseudotime, we always provide the correct root cells whose simulation times are 0 for simulations. For real dataset, we set to the cells from the cell type that is expected to be the progenitor. In Monocle 3, when the cells form disconnected clusters, only cells on the partition that includes the root cell has finite pseudotime. We only consider those cells and discard other cells with infinite pseudotime for comparison with simulation time. For slingshot, we manually set the true start cluster based on the simulation time for simulation and the cell type in real datasets. For veloVI, we use the mean latent time across genes for comparison with simulation time.

## Results

### A trajectory model generalizing cellular states

We begin by defining the trajectory as a dynamical process underlying all cells, with potentially different lineages/branches within this process. Cells are then assumed to be sampled from various, unobserved, time points along this process. Thus, the latent variable $z = (t, l)$ is introduced to account for such heterogeneity due to process time ($t$) and lineage ($l$), and $z$ follows a sampling distribution determined by the specific biological system and experimental conditions. Consequently, the probability distribution of the data we obtain is a mixture of cells over time and lineage,

$$P(\mathbf{x}|\theta) = \int_{\mathbf{z}} P(\mathbf{x}|\mathbf{z}, \theta) P(\mathbf{z}) d\mathbf{z}, \tag{1}$$

where $\mathbf{x}$ is the data and $\theta$ is the set of parameters that define the trajectory. This is the common framework of trajectory inference, and developing a trajectory model requires defining the gene dynamics along the lineage and the process time $P(\mathbf{x}|\mathbf{z}, \theta)$, as well as the sampling distribution of cells over the process $P(\mathbf{z})$.

To define the dynamical process, we first state our transcription model. We consider only transcription, splicing and degradation reactions in cells, and assume only transcription rates are time-dependent (Fig 1 Gene expression):

$$\varnothing \xrightarrow{A_l(t)} U, \; U \xrightarrow{\beta} S, \; S \xrightarrow{\gamma} \varnothing. \tag{2}$$

where $A_l(t)$ is the transcription rate function for lineage $l$ at time t, and $\beta$ and $\gamma$ are the splicing, and degradation rates. The chemical master equation describing the evolving distribution of the above biochemical reaction network has an analytical solution [23]. If we assume the initial distribution to be Poisson, the solution remains Poisson with the means ($\lambda_u$ and $\lambda_s$) of $U$ and $S$ evolving according to the following ordinary differential equations (ODEs) of RNA velocity:

$$\frac{d\lambda_u(t)}{dt} = A_l(t) - \beta\lambda_u(t),$$
$$\frac{d\lambda_s(t)}{dt} = \beta\lambda_u(t) - \gamma\lambda_s(t). \tag{3}$$

The modeling of transcription rate $A_l(t)$ is motivated by the common abstraction of cellular differentiation as cell state transitions: each lineage is abstracted as a series of switches in cellular states over time. The series of switches are specified by the given trajectory structure, which includes a directed graph of cell states where each lineage corresponds to a path (Fig 1 Input Structure). We introduce one transcription rate per gene for each cellular state ($\alpha$). Switching is assumed to be instantaneous and occurs at an unknown but fixed time ($\tau$) with the first switch leaving initial state 0 to occur at $\tau_0$. Without loss of generality, we consider the entire process to start at time 0 ($\tau_0$=0) and have a time length of 1 (e.g. $\tau_2$=1 in Fig 1). Consequently, the transcription rate function $A_l(t)$ of lineage l is simplified as piecewise constant functions of the process time over $[0, 1]$ (Fig 1 Model). This piecewise constant function is defined in the limiting regime where transcriptional state switching (such as expression of master regulatory factors and changes of chromatin state) precedes gene expression and has a much faster time scale. Thus, the piecewise constant function serves as a reasonable approximation when the time scale of transcription rate changes is comparable to or larger than the mRNA half-life. It also directly reduces to discrete cell clusters in the fast dynamic limit, i.e., when dynamical timescale ($\frac{1}{\beta}$ and $\frac{1}{\gamma}$) is much smaller than sampling intervals, for example, the total time length divided by cell number, $n$, under a uniform sampling distribution. This connection to cluster models enables us to interpolate between discrete cell states and continuous dynamics.

The simple form of the transcription rate function lead to a tractable model and facilitates inference and analysis. In fact, it affords explicit solutions for the distribution and for its derivatives with respect to parameters. Ideally, gene regulatory networks involved in cell differentiation would be modeled, but with current (transcriptomic) data types it is difficult to include gene interactions and to model transcription rates as (protein-mediated) functions of other genes with accuracy. Thus, we assume that the dynamics of different genes are independent and that all correlations are absorbed into the shared latent process time. Additionally, for simplicity, we can assume all genes are fully synchronized, as in the synchronized model, where $\tau$ is the same for all genes. However, the desynchronized model, which allows for different $\tau$ values for each gene, is also available. In summary, our trajectory model is suitable for capturing the coordinated global gene expression changes instead of the detailed gene dynamics.

After deriving an explicit distribution of in vivo counts, we turn to the measurement model. We assume simple binomial sampling of each molecule, and that the average binomial sampling probability, i.e., read depth, varies between cells but remains the same for all molecules in one cell. Then, in vitro counts remain Poisson but with means adjusted by read depth. Instead of using normalized data, we estimated read depth using the total UMI

counts of near-Poissonian genes that are not used for the inference, and incorporated it into the count distribution. Therefore, we arrive at an analytical form of the conditional probabilistic distributions $P(\mathbf{x}|\mathbf{z}, \theta)$ of counts from a dynamic process by specifying the trajectory structure, gene expression model, transcription rate functions, and scRNA-seq measurement model.

The remaining part of specifying the sampling distribution $P(\mathbf{z})$ is crucial, because it breaks the scale invariance of parameters and ensures the identifiability of the model: multiplying the transcription, splicing and degradation rates by the same constant leads to the same marginal likelihood if the sampling distribution can be changed. Ideally, the specification of the sampling distribution depends on our knowledge of the studied biological system and the experimental design. For example, for stem cells that constantly divide and differentiate, we may assume a uniform sampling distribution over $(0, 1]$ with a point mass on time 0, where the point mass represents the fraction of time that cells spend in the initial proliferative state. On the other hand, for time series data, we may assume the sampling distribution of cells are centered around their captured time points with some variances. However, in practice, since there is no obvious principled way to determine such distributions, we just assume a uniform sampling distribution over process times ($t > 0$) by default, but the weight on time point 0 and different lineages can be identifiable and be updated in the inference.

A trajectory is thus defined with the above dynamical process and sampling distributions. With the parameterized form of the probabilistic distribution of counts, we can employ the Expectation-Maximization (EM) algorithm to estimate model parameters and posterior distributions of process times and cell lineages by maximizing Evidence Lower Bound (ELBO) with either warm start or random initialization (Section Inference). The synchronized model was used by default, where all genes share the same $\tau$. Since fitting the desynchronized model from scratch is more challenging, it is recommended to begin the fitting process using the results from the synchronized model. For desynchronized models, we also introduce a penalty term proportional to the squared difference between gene-wise $\tau_k$ and the global $\tau_k$ to encourage synchronization, and the coefficient for this penalty term is set as a parameter, with a default value of 0. We tested the EM algorithm and inference on simulations generated from our trajectory model (Section Simulations). Both the synchronized and desynchronized models are identifiable, and the parameters can be recovered accurately under reasonable conditions (S2 and S3 Figs). We will elaborate on these conditions in Section Identifying failure scenarios reveals the fragility of inference.

By having an explicitly parameterized distribution of raw counts, we can easily interpret results and systematically assess the model under a more principled framework. Since parameters all have biophysical meanings, we can not only directly interpret them but also validate their accuracy by comparing them to orthogonal experiments that measure the same parameters. Furthermore, we can directly select DE genes by fold changes in transcription rates across states, after filtering genes by goodness of fit (Section Gene selection). The performance can be quantified through parameter errors, aiding in the identification of both confident and uncertain scenarios.

Not only are the parameters and results more interpretable, but we can also compare different models systematically. Regarding false positives from clustered data, we can evaluate when a trajectory model is no longer appropriate by comparing it to a cluster model using standard model selection methods like AIC (S4 Fig). Importantly, the posterior distributions and multiple minima of AIC scores can hint at the lack of continuity (S4 Fig), serving as a retrospective metric when model selection methods are compromised by unaccounted noise. We also demonstrate model selection on a disconnected trajectory, which includes both a single cluster and a bifurcation (S5 Fig). Our representation of the trajectory structure as paths on

a directed graph naturally includes this scenario. We compare the results of the true model with those of a cluster model and a connected structure. AIC and BIC can correctly identify the true structure when compared to the two incorrect models (S5 Fig).

## Demonstrating performance of Chronocell on ground-truth simulations

Since the true trajectory structure may differ from both our prior knowledge and the initial structure used, we first demonstrate the inference pipeline under a slightly incorrect trajectory. The inferred parameters provide insights into the true structure, and we then apply model selection to identify the correct model. Additionally, we include non-variable genes to assess the performance of the differential expression (DE) procedure.

We used a two-lineage trajectory and uniformly sampled 10,000 cells over lineages and process times with biological plausible parameters extracted from literature [32] (Section Simulations). 100 out of 200 genes are variable (Fig 2a). We fit under a slightly wrong assumption of the trajectory structures and assumed all genes are variable (Fig 2b). For random initialization, we initialized randomly 100 times and picked the one with highest ELBO score (Fig 2c). We also fit with warm start by initiating the fitting process from correct cell clusters grouped by true time and lineages. Both types of initialization were able to converge to the ELBO with true parameters, and random initialization yielded a slightly higher ELBO compared to warm start, with a negligible difference (Fig 2d). For the following analysis, we used the fitting results of random initialization. The posterior distributions correctly recapitulate the time and lineages of cells, with a root mean squared error (RMSE) around 0.05 for the posterior mean of process time in comparison to the true time (Fig 2e). This means that the error in time of a cell is around 5% of the total time length of the trajectory in average. However, the error is not completely uniform: as cells in the second interval are closer to the steady state, they have more spread posteriors and larger errors. Since the posteriors of each cell are accurate, the posteriors averaged over cells also resemble the empirical distribution of the true time (Fig 2f).

The parameters are also recovered accurately (Fig 2g). For variable genes, all parameters are identifiable, while for non-variable genes, only $\alpha$ and the ratio of $\beta$ to $\gamma$ are identifiable. We noticed a trend that the absolute errors of $\alpha$ scale with the square root of true values, while for $\beta$ and $\gamma$, they scale with the true values (S6a Fig). This trend also appeared in other simulations (S2b and S2c Fig). Therefore, to calculate the errors of parameters, we divide the absolute error of $\alpha$ by the square root of true values and $\beta, \gamma$ by true values, so that errors of different genes are more comparable. We refer to them as normalized errors in the text. The parameter $\alpha$ tends to be estimated with higher accuracy compared to $\beta$ and $\gamma$ (Fig 2g). This aligns with intuition because, although parameters are identifiable in our trajectory model, the Evidence Lower Bound (ELBO) is nevertheless insensitive to proportional changes in $\beta$ and $\gamma$, which have minimal impact on the phase portrait in the unspliced and spliced space of each gene. To confirm this, we used the true model to calculate the Fisher information matrix and the smallest eigenvalues of each gene (S6b Fig). The corresponding eigenvectors describe the flattest direction of the ELBO. To validate that the flattest direction primarily lies in the $\beta$ and $\gamma$ parameters, we add the corresponding eigenvectors to the true parameters after normalizing it by the square root of eigenvalues, and calculate the new ELBO with the modified parameters. Indeed the resultant changes in ELBO are indeed small and $\beta$ and $\gamma$ values were mainly varied (S6c Fig), which confirms that the $\beta$ and $\gamma$ are harder to estimate accurately (Fig 2g).

To select genes whose dynamics are well-fit by the model, we evaluate their goodness of fit by comparing the gene-wise likelihood with that of clustering models (Section Gene selection). We adopt this relative likelihood criterion because absolute likelihoods of different

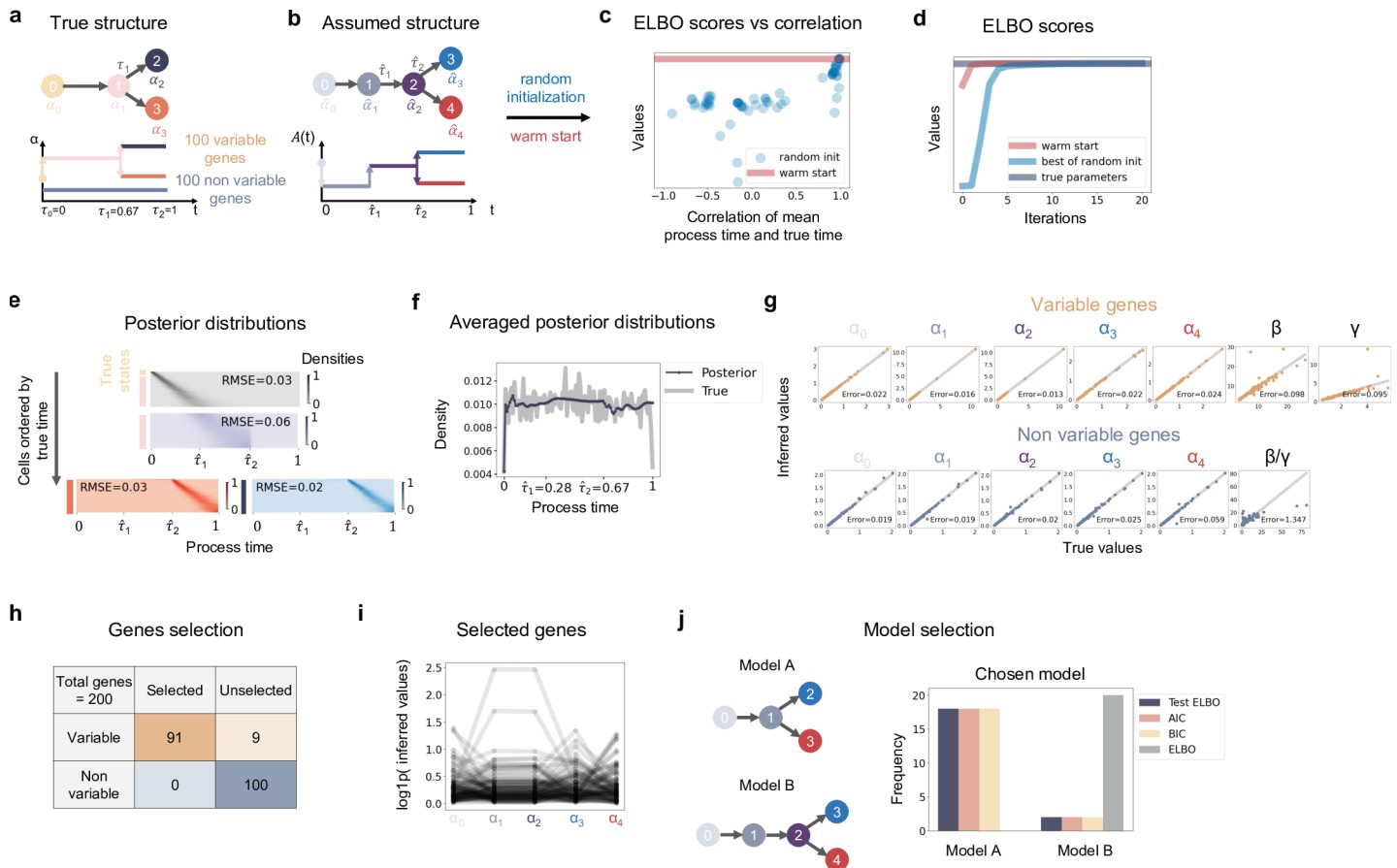

**Fig 2. Demonstration of inference on simulation. a** The ground truth trajectory structure. Cells jump to the next state (2) from starting state (1) at $\tau_0 = 0$, and then bifurcate into two lineages with different ending states (3 and 4) at $\tau_1$. The process ends at $\tau_2 = 1$. Out of 200 total genes, 100 genes are non variable with the same distributions along time. **b** The falsely assumed structure that does not know the first two states are supposed to be merged into one. All genes are assumed to vary along time. **c** The ELBO scores of 100 random initializations (blue dots) compared to those of warm start (red line). The x axis is the Pearson's correlation between the mean process time of each random initialization and the true time. **d** ELBO scores over fitting iterations of both warm start (red line) and the best random initialization (blue line), with the ELBO calculated with true parameters (gray line) as reference. **e** Heatmaps of inferred posterior distributions. x axis is time grids, and y axis is cells aligned by their true times with true transcription states on the left. The intensity of color indicates the weights of posterior distributions of cells on the grids. Heatmap of cells from $\tau_0$ to $\tau_1$ use a gray color palette. Heatmap of cells from $\tau_1$ to $\tau_2$ use purple color palette. Heatmap of cells from $\tau_2$ to $\tau_3$ of first lineage use blue, and those of second lineage use red. RMSE stands for root mean square errors. **f** The averaged posterior distributions across cells (dark blue) and true empirical distribution (gray) of process time. **g** Inferred parameters values compared to true values. For non variable genes, only $\frac{\beta}{\gamma}$ are identifiable and compared. Error is mean normalized error as described in the text, and the mean is computed across genes. **h** The confusion matrix for gene selection. **i** $\alpha$ values of selected genes over states. **j** Two models and the distribution of the chosen one by (train) ELBO, AIC, BIC and test ELBO, calculated on 20 samples each with a different set of parameter.

genes are not directly comparable and clusters model serves as a natural reference point for comparison to filter genes without continuous dynamics. All 91 genes selected belong to the variable class (Fig 2h). The similar transcription rates of states 1 and 2 of selected genes would also suggest us that those two states could be merged into a single one, if we didn't know the true structure (Fig 2i). Therefore, we applied model selection methods to compare the ground truth model and the assumed model. We generated another 20 parameter sets, fit under both models, and computed in-sample ELBO scores (ELBO), AIC, BIC, and out-of-sample ELBO scores (test ELBO). The ELBO is always better in model B due to overfitting. All of the last three metrics (AIC, BIC and test ELBO) favor the true model most of the time

(90%) (Fig 2j). However, we want to emphasize that model selection worked because simulations were strictly generated under our trajectory model. In reality, if true transcription rates are far away from piece-wise constant functions, the resulting deterministic noise can introduce bias in AIC/BIC and cross-validation, leading them to prefer more complex models [33].

## Identifying failure scenarios reveals the fragility of inference

Given the accuracy on perfect data, we sought to characterize the impact of different factors on inference accuracy and identify potential failure scenarios that could lead to unreliable results. We first study the requirements on some obvious factors like the number of cells, the number of genes, and the means of counts. Relatively small numbers of cells and genes appear to be sufficient (S7 and S8 Figs). As the number of cells increases, parameter errors decrease, while time errors remain the same. On the other hand, time errors decrease with an increasing number of genes while parameter errors remain the same. Additionally, counts means must be sufficiently high to enable accurate inference, which necessitates adequate sequencing depth (S9 Fig). We notice that when the trajectory structure becomes more complex, the requirement for count means also increases: Chronocell needs higher means for similar accuracy (S10 Fig). Furthermore, when count means are low, increasing the number of cells can actually compromise the accuracy, underscoring the critical importance of obtaining sufficient counts.

As we use piecewise-constant functions to approximate transcription rates, we also test how this simplification impacts results when transcription rates are, in fact, not piecewise-constant. We generate simulations using piecewise-exponential functions for transcription rates, which ranges from almost linear to almost piecewise as the rate constants increase. We fit the model with Chronocell under the piecewise-constant assumption, and the inference accuracy remains satisfactory when the rate constants are comparable to or larger than the mRNA half-life (S11 Fig).

As unaccounted noise is prevalent in scRNA-seq datasets, we then characterize the effects of noise on inference. The first type of noise is cell-wise read depth (or cell size) that influences all genes similarly. The inference is sensitive to such noise: when read depth variance is not correctly accounted for in the fitting, inference results may be highly inaccurate when its squared coefficient of variation ($CV^2$) exceeds 0.1 (S12a Fig). As the $CV^2$ typically observed in real datasets exceeds 0.1 (S17 Fig), an accurate estimate of cellwise read depth is critical. Considering this fact, instead of simply using total counts as normalization factors, we use normalized covariance between genes to decompose extrinsic noise (influencing all genes) and intrinsic noise (gene specific). We estimate the read depth $CV^2$ across cells from the normalized covariance between genes. Subsequently based on the read depth $CV^2$, we subtract the extrinsic variance caused by the read depth from total variance and identify Poissonian genes whose remaining variances (intrinsic variances) are close to their means (variance < 1.2 mean). We then estimate cell-wise read depth using the sum of those Poissonian genes. Different sets of genes are used for estimating the $CV^2$ of read depth and fitting trajectories. In reality, these read depth estimates correlate well with total counts number for most datasets, with one interesting exception (S17 Fig).

On the other hand, gene specific noise seems to have less impact on inference. We added gene-wise gamma noise in simulation which generates negative binomial distributions in steady states and approximates bursting noise. Parameter errors gradually increase as $CV^2$ of Gamma noise increases but remain reasonably small even with a $CV^2$ of 1 (S12b Fig). However, while it may not completely undermine the fitting process, it can lead to failures

in model selection. When repeating the model selection procedures in Figs S4 and 2 after introducing Gamma noise, the AIC/BIC and cross-validation metrics favor the wrong models that are more complex than the true one (S12c and S12d Fig). This highlights the importance of providing reasonable trajectory structure to the fitting based on prior knowledge, and exploring alternative methods for quality control against false positives caused by clusters.

Another probable factor in real datasets that can lead to suboptimal results is insufficient dynamics. Intuitively, for a dynamical model to be appropriately fit, the data must capture a significant amount of transient dynamics to recapitulate the evolving processes over time, without which the data start to resemble discrete clusters. Such cases can occur in at least two possible scenarios: fast timescales and concentrated sampling distributions, both of which result in clusters in the extreme. Therefore, false positives caused by clusters are naturally included as a component of identifying unreliable results.

The first situation, fast timescales, arises when mRNA half-lives are significantly shorter than the timescale of biological processes, thus cells are mostly near steady state and provide little information about the intermediate dynamics. By setting the length of the time interval to unity and increasing $\beta$ and $\gamma$, which is equivalent to increasing processes timescale with unchanged mRNA half-lives, we found that ideally, the mean value of $\gamma$ should not fall out of the range of 1 to 10, and the mean ratio of $\gamma$ to $\beta$ over genes should not be too small (S13 Fig). Hence, if the average half-life of spliced mRNA is approximately 30 minutes [32], snapshot data sampled from processes involving steps exceeding 10 hours are no longer suitable.

Furthermore, given an appropriate timescale, the sampling distribution still has to cover the region where the transient dynamics occur. Imagine, for example, that all cells are from one unknown time point; there is no way for a trajectory to be inferred. Instead, a cluster should be used to fit the data. Thus, it is crucial to determine the minimum level of uniformity required in the sampling distribution and verify whether this requirement is met. By gradually changing sampling distributions from a uniform distribution to a Gaussian with a random mean, we generated datasets with sampling distributions that exhibit decreasing levels of uniformity, which was quantified using entropy (S14a Fig). As the sampling distribution deviates from uniformity, the errors of process time and parameters quickly increase as expected (S14b Fig). In fact, even when the sampling distribution is not far away from a uniform one with entropy 4, the errors are big enough that the results are completely wrong (S14b Fig). Further, even using the true sampling distribution as a prior for a warm start with correct initialized time cannot mitigate the lack of dynamics: the errors in parameter estimations remained substantial (S14c Fig). Therefore, sufficiently transient dynamics is an inherent requirement for trajectory inference even when perfect prior information is provided.

In summary, a suitable dataset for fitting process time needs to contain enough dynamic information as well as limited noise. This stringent requirement for a successful trajectory inference highlights the need for suitable datasets and careful model assessment. Straightforwardly, we could make a consistency check by verifying if the remaining noise, parameter values, and uniformity of the average posterior distribution fall into the appropriate range determined in simulations. However, when a result is incorrect, it may not necessarily exhibit large unexplained noise, high splicing/degradation rates, or a concentrated cellular distribution over process time, as it could inadvertently fit undesired patterns and output plausible results. Thus, inconsistency is a sufficient but not necessary indicator of unreliable results.

It turns out that the large uncertainty of results can be a better indicator. As both noise and limited dynamics tend to diminish or obscure the difference of scores like ELBO between correct and incorrect outcomes, they introduce multiple comparable maxima and make the fitting results unstable. The resultant large uncertainty can be measured by two different approaches. First, as each random initialization outputs a different ELBO/AIC score and cell

ordering, we can inspect the distribution of scores with respect to a summary parameter of cell orderings, which describes the global landscape of the score function. Ideal scenarios usually give one distinct maximum of ELBO (or minimum of AIC) at the correct ordering of cells (correlation around one), while failure scenarios usually lead to multiple comparable maxima of ELBO (or minima of AIC) at different cell orderings beside the correct one (S15b Fig). We use average precision (AP) of the 100 random initializations to quantify the distinctiveness of the correct outcomes and summarize the uncertainty. One random initialization is considered correct if its resultant mean process time correlates well with that of the best one, e.g., a Pearson's correlation of 0.8 (Section Uncertainty assessment). Ideal cases lead to AP close to one and low AP indicates instability but not vice versa, which means low AP is a sufficient indicator for instability (S15b Fig). Second, the uncertainty can also be revealed by standard bootstrap analysis. We generated 100 sets of resampled data and computed the correlation in process time between the original and each resampled set (Section Uncertainty assessment). In failure scenarios the results of original data and resampled data often differ and correlations are scattered with large variance. On the other hand, in ideal scenarios, process time estimates of resampled results agree with the original one and correlations are centered around one (S15c Fig).

## Uncovering distinct underlying cellular distributions in process time

We applied Chronocell to a variety of datasets with different anticipated sampling distributions over time. For those datasets, we estimated read depth as described in Section Read depth estimation (S17 Fig), and then filtered genes for fitting based on their means, variances and unspliced to spliced ratios (Section Real datasets preprocessing). The trajectory structures are determined based on prior knowledge. Random initialization was always performed and its uncertainty was assessed by both AP and bootstrapping. Warm start was applied as well if cell type annotations were available, which were used to initialize the fitting process. The corresponding clusters (Poisson mixtures) model was fit with the same read depth and used for comparison. The genes were selected based on goodness of fit, in the same manner as in the simulation (Section Gene selection). Then, DE genes are selected based on the fold changes in transcription rates across states.

We first tested our method on the T cells of PBMC (Peripheral Blood Mononuclear Cells) dataset from 10x Genomics, which is typically expected to exhibit a few distinct clusters (S18a Fig). Indeed, though the AIC of trajectory model is lower than clusters model likely due to unaccounted noise, the scores of 100 random initializations display multiple minima: multiple different cell orderings result in similar AIC values (S18b Fig). The low average precision (0.28) of random initializations suggests clusters model would be more suitable for PBMC. Serving as a negative control, it confirms our ability to reject unreliable results on real datasets, even when standard model selection methods are invalidated by incorrect modeling of noise.

The second dataset contains a snapshot collection of glutamatergic neuronal lineage cells in a developing human forebrain (Fig 3a) [16], which is presumed to capture cells along a continuous trajectory. However, the unstable result of random initializations indicates its unreliability, as results with reversed directions yield comparable AIC scores (Fig 3b), resembling simulations that lack dynamics information (S15b Fig). This observation is further supported by examining the cellular posterior distributions obtained through warm start with cell type annotations, where the average posterior distribution reveals that cells are concentrated around starting time $\tau_0$, i.e., the initial state, with a low entropy (Fig 3c). Therefore, both the

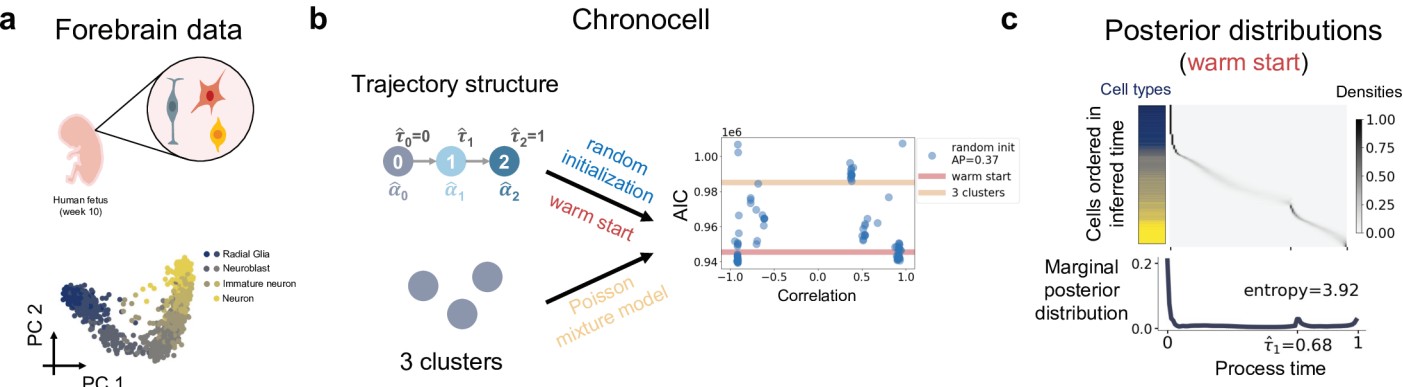

**Fig 3. Inference results for Forebrain data. a** Schematics of Forebrain data and PCA plot of cells colored by cell type annotations. **b** The AIC scores of the trajectory and cluster models. The x axis is the mean process time correlations of 100 random initializations (blue dots). The AIC scores of random initializations are compared to those of warm start (red line) as well as 3 Poisson mixture model (yellow line). AP stands for average precision. **c** Posterior distributions of process time of the trajectory model with warm start. Cells were ordered in y axis by their inferred mean process time and the left bar displays their cell types using the colors in **a**. The below histogram shows average posterior distribution averaged over cells. The entropy of the average posterior distribution was calculated using its weights on the 100 discretized time grids.

inconsistency indicated by the concentrated posterior and instability indicated by comparable peaks of AIC suggest this is not a suitable dataset for Chronocell and likely lacks enough dynamics.

The third dataset contains erythroid lineage cells during mouse gastrulation collected from multiple time points (Fig 4a) [34]. Biological time availability offers a valuable means to evaluate results by comparing the posterior distributions of cells to their corresponding physical times, which is particularly useful since real snapshot datasets lack a definitive ground truth. The AIC scores of random initializations show a clear minimum at a correct direction that align with cell type annotations (S19a Fig), and the average precision of random initializations is 0.79 which is notably higher than those of PBMC and Forebrain datasets (Figs S18b and 3b). The process times of bootstrap samples are reasonably stable as well (S19a Fig). The fit dynamics were able to explain most of the variance, leaving the $CV^2$ of unexplained noise of most genes under 1 (S19c Fig). Therefore, the Erythroid dataset appears to be a suitable dataset for our trajectory model. The best result from random initialization is used for analysis (Fig 4b), and cellular posterior distributions confirm that cells have a broad distribution over process time (Fig 4c). Furthermore, the posterior distributions of cells do not differ significantly until E7.5, after which they roughly progress along the process time in sync with real-time progression (Fig 4d). This observation aligns with the understanding that erythroid differentiation in a mouse embryo is believed to begin around embryonic day 7.5 (E7.5) [35]. Although the alignment with physical time is not perfect, it still suggests that our trajectory model can successfully capture the correct trend of process time, even when assuming a uniform sampling distribution for all cells.

After applying our gene selection procedure based on relative likelihood and discarding genes with extreme values, we ended up with 24 (49%) genes (Fig 4e). Subsequently for demonstration, we chose the top 5 DE genes (*Cpox, Smim1, Abcg2, Rbpms, Prtg*) with the largest fold change of transcription rates, and plotted their phase portraits (Fig 4f). Interestingly, four of them (*Cpox* [36], *Smim1* [37], *Abcg2* [38], *Rbpms* [39]) were reported to be directly relevant to erythroid development, which illustrates that selecting DE genes based on inferred transcription rates is a straightforward and effective approach.

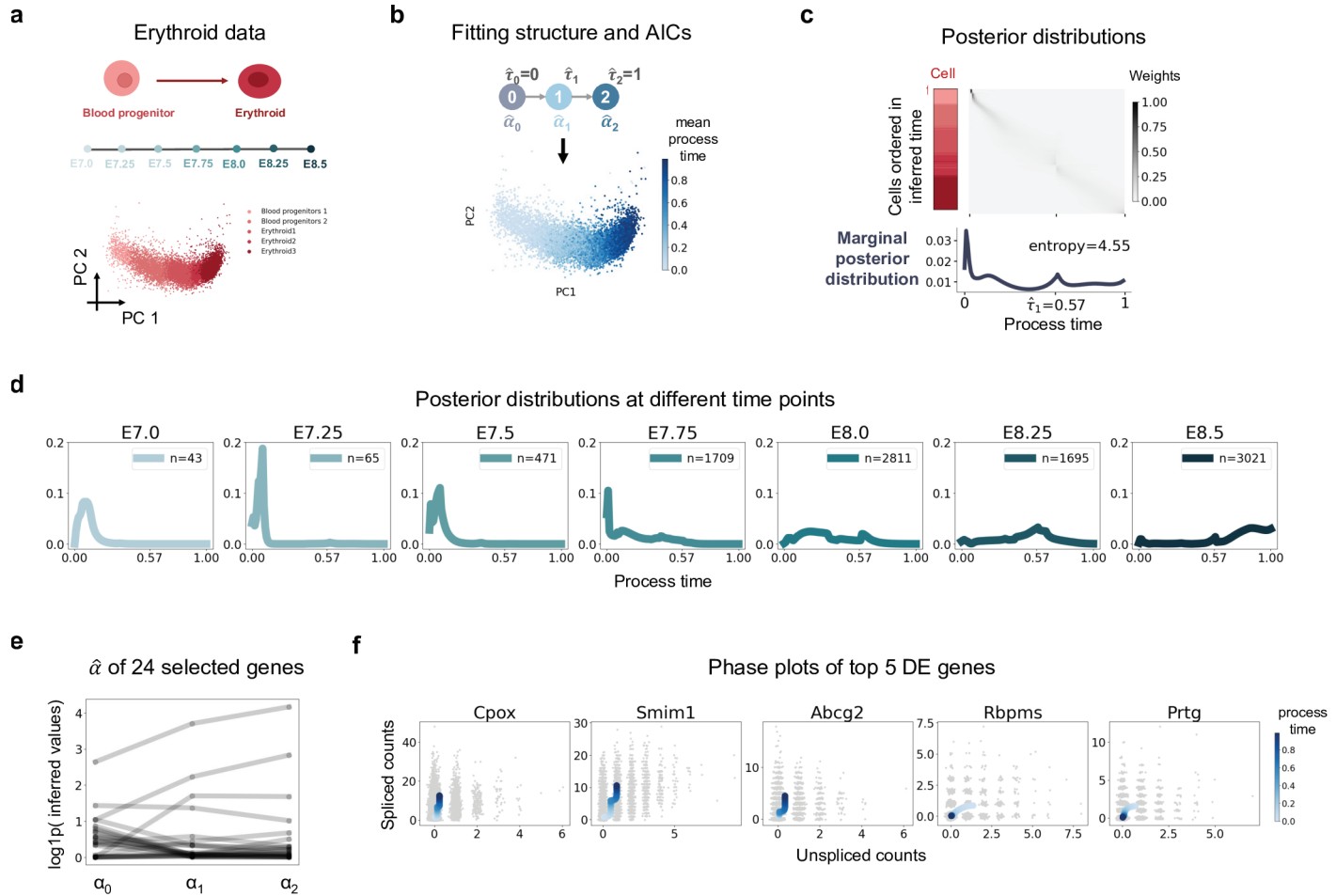

**Fig 4**. **Inference results for Erythroid data**. **a** Schematics of Erythroid data and PCA plot of cells colored by cell type annotations. **b** The fit trajectory structure and inferred mean process time from random initialization indicated in blue on the same PCA plot as in **a**. **c** Posterior distributions of process time. Cells were ordered in y axis by their inferred mean process time and the left bar displays their cell types using the colors in **a**. The below histogram shows average posterior distribution over cells. The entropy of the average posterior distribution was calculated using its weights on the 100 discretized time grids. **d** Averaged posterior distribution across cells from different experimental time points. n is the number of cells. **e** α values of 24 selected genes over states. **f** Phase plots of top 5 DE genes of 24 selected genes. The x axis is the raw unspliced counts and y axis is the raw spliced counts. The blue curve is the fit mean of product Poisson distributions of unspliced and spliced counts over process time, and its darkness corresponds to the value of process time.

Based on the results from datasets ranging from clusters to trajectories, it becomes evident that real datasets can display a spectrum of continuity in cellular process time distribution. Therefore, it is critical to assess the quality of inference and verify the requirements for reliable results are indeed met.

## Degradation rates estimates agree with metabolic labeling data

In addition to validating the process time, we also sought to validate the inferred parameters, which motivated us to use a metabolic labeling dataset from scEU-seq, comprising human retinal pigment epithelial (RPE1) cells undergoing cell cycles (Fig 5a) [40]. Metabolic labeling of new mRNA allows for the estimation of degradation rates from cells with varying labeling times, enabling a comparison with our parameter estimations.

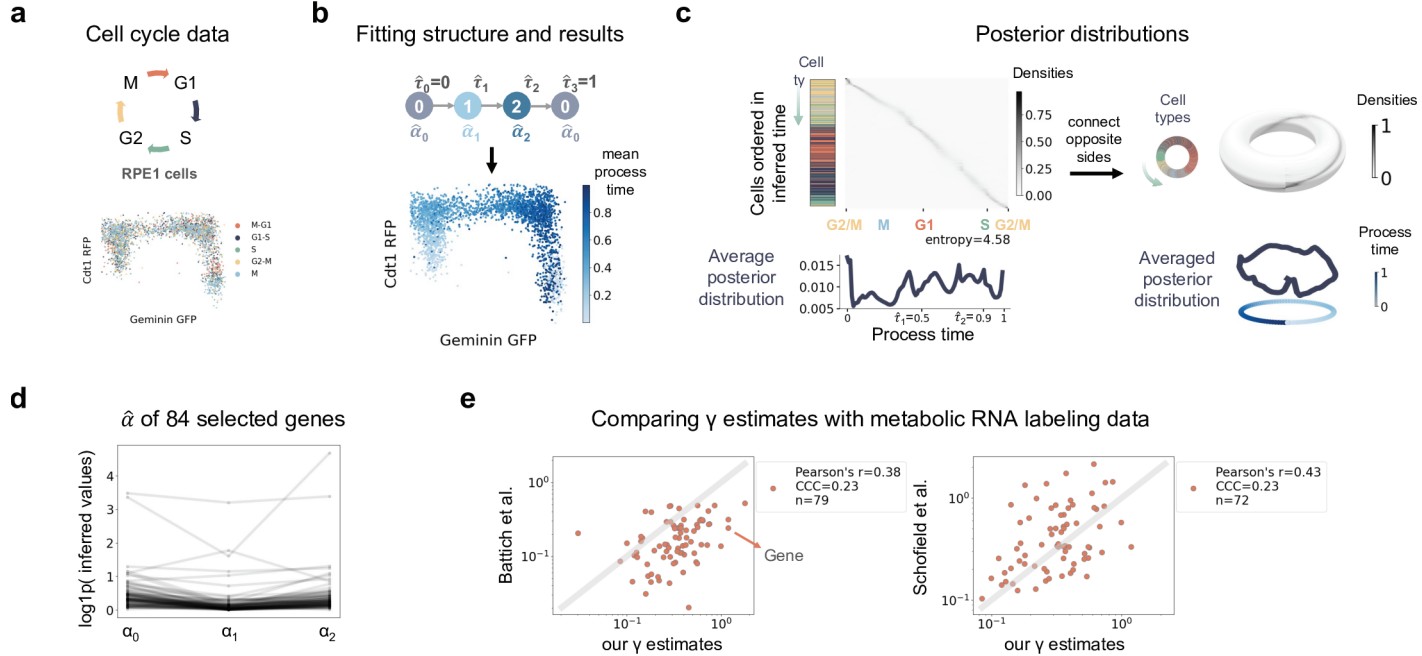

**Fig 5. Inference results for cell cycle data. a** Schematics of Cell cycle data and scatter plot of the Geminin-GFP and Cdt1-RFP of RPE1 cells colored by cell type annotations. **b** The fit trajectory structure and inferred mean process time from random initialization indicated in blue on the same scatter plot as in **a**. **c** Posterior distributions of process time. Cells were ordered in y axis by their inferred mean process time and the left bar displays their cell types using the colors in **a**. The below histogram shows average posterior distribution averaged over cells. The entropy of the average posterior distribution was calculated using its weights on the 100 discretized time grids. **d** $\alpha$ values of 84 selected genes over states. **e** Comparison of $\gamma$ estimates for 84 selected genes with estimates derived from metabolic RNA labeling data. CCC stands for concordance correlation coefficient. n is the number of genes for which estimates are available in each respective paper.

In the trajectory structure, we specified that the initial and final states were identical. Given our assumption that the initial state is at a steady state, capturing the cyclic nature of the cell cycle poses an additional requirement that cells must also approach a steady state in the last interval as well, which happened to be reasonably satisfied by our results. In line with the expectation that cell cycle is a highly dynamic process, the random initialization provides a relatively stable result: the AIC exhibits a single minimum that aligns with the direction of cell cycle progression (S20a Fig), and bootstrap samples mostly align with the result of the original one (S20b Fig). Starting from the result of random initialization, a desynchronized model was fit to enhance the accuracy of parameter estimations (S20c Fig). The resulting fit maintains alignment with the cell cycle progression (Fig 5b), and the $CV^2$ of the unexplained noise mostly remains below 1 for most of the genes (S20c Fig). The RPE1 dataset was metabolically labeled for different lengths of time but posterior distributions of RPE1 cells with different labeling times do not show significant differences, as a negative control in contrast to the erythroid data (S20e Fig).

The posterior distributions successfully capture the cyclic nature of the data. The cell types ordered by mean process time exhibit a cyclic pattern (Fig 5c); the phase plots of marker genes confirm the cycling dynamics of fit means of unspliced and spliced counts (S20f Fig); the starting and ending values of fit means of most of genes match with each other (S20g Fig). As both axes are cyclic, it is possible to connect the opposite edges and transform the two-dimensional cell-by-time grids posterior distributions into a torus on the surface of which the posterior distributions look like a circle (Fig 5c). Based on cell type annotation, total RNA

counts number (S20h Fig), and marker genes dynamics (S20e Fig), we can roughly assign the three intervals to G2/M, G1/S and S/G2 phase, and mitosis happens shortly after $\tau_0$.

After selecting genes by relative likelihood and discarding genes with extreme values, we ended up with 84 (46%) genes (Fig 5d). We compared the degradation rates of 84 selected genes to those derived by scEU-seq [40] and TimeLapse-seq [41]. For scEU-seq, as we neglected the changes in degradation rate along cell cycle, we used the averaged degradation rates in S10c Fig of Battich et al. [40]. We observed moderate correlations between our degradation rate estimates and the respective estimates from Battich et al. and Schofield et al. (Fig 5e), and these correlations exhibited a magnitude similar to the correlation between the Battich et al. and Schofield et al. estimates (S20i Fig), which suggests that the parameters inferred by Chronocell indeed possess a meaningful biophysical interpretation.

## Discussion

We have introduced Chronocell, a method constructed upon a trajectory model featuring biophysically meaningful parameters and a principled approach to fitting and analysis. Counts are directly modeled, estimation accuracy is characterized, and unreliable instances are discerned. We found several requirements regarding dynamics and noise that must be satisfied to obtain reliable results, which makes process time inference challenging and renders retrospective model assessment indispensable. We applied Chronocell to different kinds of real datasets, recognized inapplicable ones by assessing instability, and demonstrated meaningful interpretation of process time and parameters on applicable ones.

Of note, our trajectory model is simplified to balance interpretability and tractability. Building on the concept of cell state transitions, we have assumed piecewise constant transcription rates which describe genes with fast chromatin states switching in saturated regime, but may be unrealistic for many biological systems. Furthermore, we have assumed that splicing and degradation rates remain constant, which although reasonable for many genes, is not accurate for genes with peaked response [32]. We also did not incorporate transcriptional bursting due to the absence of an analytically determined temporal solution, and this leads to a under-dispersed distribution compared to what is typically observed in biological data. Nevertheless, this can be resolved by using numerical solutions [42]. However, even with this simplified model, we have found that accurate inference imposes strict requirements on data. This question is inherently challenging and insufficiently specified due to the existence of latent variables and flexibility of the transcription rates. A more realistic model would have even more stringent requirements.

Since fitting dynamical parameters is challenging from static snapshots, physical time information could offer valuable insights when incorporated into our latent variables model. While the requirements on sampling and noise might potentially be relaxed, they would likely persist to some extent. Thus, intensely sampled time series datasets derived from well-defined cellular processes would be the ideal choice for trajectory inference. However, it remains to address the key question of how physical time should be translated into sampling distribution assumptions. The choice of sampling distribution reflects our understanding of cell heterogeneity at each time point. Should we fix the process time to physical time completely, i.e., a delta distribution? Or should we assume heterogeneity within one time point? If so, what kind of distribution should we use? There is no straightforward answer, as it depends on our understanding of the process being studied. For instance, in a neuron dataset with cells sampled at five time points after stimulation, assuming homogeneous cell responses would suggest using a delta distribution, which aligns process time directly with physical time. However, if we account for heterogeneity within a time point, other distributions may be more

appropriate. Furthermore, assuming different distributions (delta, exponential, uniform) can lead to different results (S21 Fig). This underscores the necessity for a more comprehensive understanding and modeling of cell heterogeneity, and optimal experimental design for both the number and timing of the time points to generate more informative data.

We have not yet provided a benchmark against other methods. Each trajectory inference method assumes a different model, and comparing methods with different underlying models is often less informative without ground truth. For example, descriptive models use conceptually different approaches, so comparing them is an "apples to oranges" situation, as they can not recover the true time of data under Chronocell trajectory model. On the other hand, they may be more suitable for exploratory analysis, with fewer data requirements and being less computation-demanding. Similarly, in model-based approaches, the model inherently reflects our understanding of the data, and discrepancies in results arise from differences between models. This underscores the importance of using models that are grounded in biophysical motivation. For demonstration purposes, we provide a comparison of Chronocell with some widely used trajectory inference methods, including Slingshot [9], Monocle 3 [6], and diffusion pseudotime [43], as well as recently published veloVI which also integrates trajectory inference with RNA velocity [13]. These comparisons are demonstrated on the simulated data used for illustration in Fig 2, as well as simulations generated by dyngen [24] (S22 Fig). We find that other methods can not correctly recover true time on data generated under the Chronocell model (S23 Fig). For dyngen simulation, Chronocell has comparable or better accuracy though all methods capture the correct trend but fail to recover true time accurately. For real data, benchmarking is more challenging due to the lack of ground truth. The closest approximation to ground truth is the experimental time in time series data, especially with short enough intervals and a clear start point. Despite the uncertainty of assumed cell heterogeneity discussed above, at least the inferred time of cells should proceed along with experimental time. Therefore, we also test other methods on the neuron data and find that only Monocle 3 captures the general trend of time progression (S24 Fig).

In summary, our biophysically motivated model of the dynamical processes captured in single-cell data enables the inference of process times and parameters with biophysical interpretations. It presents an alternative approach to unveil continuous latent cell representations within a well-defined and rigorous framework, and highlights the limitations of what can be inferred using current snapshot single-cell genomics data.

## Supporting information

**S1 Text. Supplementary notes.**
(PDF)

**S1 Fig. False positive on clusters data**. **a)** Negative control data are simulated from 4 Poisson mixtures with read depth noise. **b)** As an example of false positive, specious trajectory in lower dimensional space was constructed with Slingshot [9]. **c)** Differential genes along pseudotime were selected with tradeSeq [44], with the first gene plotted along the blue lineage.
(PDF)

**S2 Fig. Inference accuracy of synchronized model**. **a)** The two trajectory structures used in simulations. **b)** Estimation errors of different parameter sets. For time, error is root mean square error. For $\alpha, \beta, \gamma$, error is mean normalized error as described in the Section Simulations. **c)** Absolute errors with respect to the true values of parameters.
(PDF)

**S3 Fig. Inference accuracy of desynchronized model**. Estimation errors of the desynchronized model tested on simulations with trajectory structures in S2a Fig. **a)** Errors of different

parameter sets. For time, error is root mean square error. For $\alpha, \beta, \gamma$, error is mean normalized error as described in the Section Simulations. **b**) Absolute errors with respect to the true values of parameters.
(PDF)

**S4 Fig. Inference and model selection on clusters data. a)** The same data from 4 Poisson mixtures as in S1 Fig. **b**) To compare the trajectory and Poisson mixture models, AIC scores of Poisson mixtures model and trajectory model are compared on 20 simulations with different random parameter sets. Dots below y=x indicate Poisson mixture model is better.
(PDF)

**S5 Fig. Inference and model selection on disconnected data. a** The ground truth trajectory structure. A subset of cells is from a bifurcation trajectory and the other cells are from a disjoint cluster. For the bifurcation trajectory, cells start from the state 0 and jump to the state 1 at $\tau_0 = 0$, and then bifurcate into two lineages with different ending states (2 and 3) at $\tau_1 = 0.5$. The process ends at $\tau_2 = 1$. **b** Heatmaps of the inferred posterior distributions for cells from both the bifurcation and the cluster. x axis is time grids, and y axis is cells aligned by their true times and grouped by their true lineages. The intensity of color indicates the weights of posterior distributions of cells on the grids. Heatmap of cells from $\tau_0$ to $\tau_1$ use a purple color palette. Heatmap of cells from $\tau_1$ to $\tau_2$ of first lineage use blue, and those of second lineage use red. Heatmap of cells from cluster use an orange color palette. RMSE stands for root mean square errors. **c** Inferred parameters values compared to true values. Error is mean normalized error across genes as described in the Section Simulations. **d** AIC and BIC of the true model and two wrong models.
(PDF)

**S6 Fig. Supplementary figures for demonstration of inference on simulation. a)** Absolute errors with respect to the true values of parameters. **b**) Distribution of the smallest eigenvalues of the Fisher information matrix of each gene. **c**) Marginal likelihood (ELBO) and varied parameters compared to true parameters. The difference between varied parameters and true parameters are the eigenvectors corresponding to the smallest eigenvalues of the Fisher information matrix, divided by the square root of the respective eigenvalues, specific to variable genes.
(PDF)

**S7 Fig. Impact of cell numbers on inference accuracy and running time. a)** Estimation errors for datasets with varying cell numbers. The trajectory structures are the same as in S2a Fig. For time, error is root mean square error. For $\alpha, \beta, \gamma$, error is mean normalized error as described in the Section Simulations. Estimation errors of different cell numbers. **b**) Running time of 100 epochs on a single core on datasets with varying cell numbers.
(PDF)

**S8 Fig. Impact of gene numbers on inference accuracy**. The trajectory structures are the same as in S2a Fig. For time, error is root mean square error. For $\alpha, \beta, \gamma$, error is mean normalized error as described in the Section Simulations. **a**) Results for trajectory structure 1. **b**) Results for trajectory structure 2.
(PDF)

**S9 Fig. Impact of mean counts on inference accuracy**. Simulations of different counts mean are generated by scaling the transcription rates while keeping other parameters the same. The trajectory structures are the same as in S2a Fig. For time, error is root mean square error. For $\alpha, \beta, \gamma$, error is mean normalized error as described in the Section Simulations. **a**) Results for trajectory structure 1. **b**) Results for trajectory structure 2.
(PDF)

**S10 Fig. Impact of trajectory structure complexity on inference accuracy**. **a**) The trajectory structure used in this figure. **b**) Impact of counts means on inference accuracy. Datasets with increasing counts means are generated by increasing the mean parameters $\mu$ in log-normal distributions for $\alpha$. **c**) Results on 10 random parameter sets with different distributions for $\alpha$. **d**) Impact of cell numbers on inference accuracy on simulations with different distributions for $\alpha$.
(PDF)

**S11 Fig. Impact of non piecewise-constant transcription rate functions on inference accuracy**. Simulations are generated using piecewise-exponential functions for transcription rates and fit under Chronocell's piecewise-constant assumption. The three trajectory structures have been defined in S2a and S10a Figs. For time, error is root mean square error. For $\alpha, \beta, \gamma$, error is mean normalized error as described in the Section Simulations. **a**) One piece of the piecewise-exponential functions used for transcription rates. Different rate constants are used in the simulation to span the range from an almost linear transition to an almost step function. **b**) Results for trajectory structure 1. **c**) Results for trajectory structure 2. **d**) Results for trajectory structure 3.
(PDF)

**S12 Fig. Impact of noise on inference accuracy and model selection**. The trajectory structures are the same as in S2a Fig. For time, error is root mean square error. For $\alpha, \beta, \gamma$, error is mean normalized error as described in the Section Simulations. **a**) Estimation errors as read depth noise increases. **b**) Estimation errors as gene-wise Gamma noise increases. **c**) Impact of gene-wise Gamma noise on model selection on clusters data. Same as in S4 Fig except Gamma noise with $CV^2 = 1$ was added to Poisson mixtures to generate simulation data (Section Simulations). **d**) Impact of gene-wise Gamma noise on model selection of trajectory structure. Same as in Fig 2j except Gamma noise with $CV^2 = 1$ was added in simulation.
(PDF)

**S13 Fig. Impact of dynamic timescale on inference accuracy**. The trajectory structures are the same as in S2a Fig. For time, error is root mean square error. For $\alpha, \beta, \gamma$, error is mean normalized error as described in the Section Simulations. **a**) **i** Schematics of phase plots with increasing timescale. **ii** Estimation errors as time scale increases. **b**) **i** Schematics of phase plots with the ratio of $\gamma$ to $\beta$ increasing while keeping their product constant. **ii** Estimation errors as $\frac{\gamma}{\beta}$ increases.
(PDF)

**S14 Fig. Impact of sampling distribution uniformity on inference accuracy**. The trajectory structures are the same as in S2a Fig. For time, error is root mean square error. For $\alpha, \beta, \gamma$, error is mean normalized error as described in the Section Simulations. **a**) Schematics of sampling distributions used in simulation with decreasing uniformity. The sampling distributions were gradually changing from uniform distribution to Gaussian distribution. Right plot shows the entropy of the sampling distributions. **b**) Estimation errors as uniformity decreases under uniform prior. **c**) Estimation errors as uniformity decreases warm started with correct position under true prior. Fitting was initialized with posteriors calculated under true parameters, and empirical distribution of process time of samples were provided as prior for the sampling distribution.
(PDF)

**S15 Fig. Uncertainty and lack of robustness as an indicator of failure scenarios**. **a**) Schematics of three probable failure factors: clusters data, fast timescale, concentrated sampling distribution. Two example simulations were used for each case and the gray arrows indicate the their difference. **i** The two simulations are the cluster data in S4 Fig and noisy cluster

data in S12c Fig respectively. **ii** The two simulations are the 5th and 13th instances of structure 1 in S13b Fig. **iii** The two simulations are the 1st and 11th instances of structure 2 in S14b Fig. **b)** AIC vs. correlation of mean process time of 100 random initializations in different scenarios. Results of two example simulations were showed. The x axis is the correlation of mean process time between each initialization and the best one. The gray arrows correspond to those in **a. c)** AIC vs correlation of mean process time of 20 bootstrap samples in different scenarios. Results of the two example simulations were presented in the same position as in **b**. The x axis is the correlation of mean process time between each bootstrap sample and the original one (which is the best one in **b**). The gray arrows correspond to those in **a**.
(PDF)

**S16 Fig.** $CV^2$**-mean relationship of total, unspliced and spliced counts**. **a)** Forebrain data. **b)** Erythroid data. **c)** Cell cycle data. **d)** Neuron data. **e)** PBMC data.
(PDF)

**S17 Fig. Read depth estimation using normalized covariance for highly variable gene selection**. **a)** Read depth estimation based on total counts of Poissonion genes. Cells are colored by their cell types. **b)** Selected genes (black) for fitting plotted in the same $CV^2$-mean plot as in S16 Fig.
(PDF)

**S18 Fig. Inference results for T cells from PBMC data**. **a)** Schematics of T cells from PBMC dataset and PCA plots. **b)** The fit trajectory structure and AIC scores of 100 random initializations (blue dots) compared to those of 3 clusters (Poisson mixtures) model (yellow line). AP stands for average precision.
(PDF)

**S19 Fig. Supplementary figures for Erythroid data**. **a)** AIC scores and mean process time correlations of 100 random initializations (blue dots) compared to those of warm start (red line) as well as 3 clusters (Poisson mixtures) model (yellow line). AP stands for average precision. Mean process time of the initialization with lowest AIC is indicated in blue on the same PCA plot as in **a**. **b)** AIC scores and mean process time correlations of 100 bootstrap samples. The x axis is the Pearson's correlation between the mean process time of each bootstrap and the those of original data, i.e., the plotted one in **a**. **c)** Distribution of remaining squared coefficient of variance of 49 genes used in the fitting. Remaining squared coefficient of variance is calculated by dividing the remaining unexplained variance by mean squared.
(PDF)

**S20 Fig. Supplementary figures for Cell cycle data**. **a)** AIC scores and mean process time correlations of 100 random initializations (blue dots) compared to those of warm start (red line) as well as 3 clusters (Poisson mixtures) model (yellow line). AP stands for average precision. Mean process time of the initialization with lowest AIC is indicated in blue on the same PCA plot as in **a**. **b)** AIC scores and mean process time correlations of 100 bootstrap samples. The x axis is the Pearson's correlation between the mean process time of each bootstrap and the those of original data, i.e., the plotted one in **a**. **c)** ELBO scores over iterations for desynchronized model. The fitting started with the best random initializations result of synchronized model. **d)** Distribution of remaining squared coefficient of variance of 182 genes used in the fitting. Remaining squared coefficient of variance is calculated by dividing the remaining unexplained variance by mean squared. **e)** Averaged posterior distribution across cells with different labeling times. n is the number of cells. **f)** Dynamics of three marker genes. The blue curve is the fit mean of product Poisson distributions of unspliced and spliced counts over process time, and its darkness corresponds to the value of process time. Cells' raw counts (gray) are plotted against their corresponding process times. **g)** Starting and ending values

of fit mean of Poisson distributions. **h**) Total counts over process time of cells colored by cell type annotations. **i**) Comparison of $\gamma$ estimates from two metabolic RNA labeling papers for 84 selected genes. Estimates of 67 genes are available in both papers. CCC stands for concordance correlation coefficient.

(PDF)

**S21 Fig. Inference results of different sampling distribution assumption for Neuron data**. Fitting was warm started from delta distribution at physical time under different sampling distribution priors using the shown trajectory structure. **a**) The assumed sampling distribution. For **iii**, uniform distribution is assumed for cells from all time points. **b**) The fit trajectory structure and inferred mean process time indicated in blue on the PCA plot. **c**) Violin plots of mean process time of cells with different labeling times. Three blue bars represent the mean and extremes.

(PDF)

**S22 Fig. Results of Chronocell compared to other methods on simulations generated by dyngen**. Chronocell, Monocle 3 [6], Slingshot [9], diffusion pseudotime [43] and veloVI [13] are applied on simulation generated using dyngen [24]. Inferred time is plotted against true time, where x axis is the true simulation time normalized between 0 and 1 and y axis is corresponding inferred time normalized between 0 and 1. RMSE stands for root mean square error of inferred time. **a**) The dyngen simulation projected into the first two principal component spaces. A bifurcation backbone is used. **b**) The fit trajectory structure and results of Chronocell. **c**) The results of Monocle 3. **d**) The results of Slingshot. **e**) The results of diffusion pseudotime. **f**) The results of veloVI.

(PDF)

**S23 Fig. Results of other methods on Fig 2 simulation**. Monocle 3 [6], Slingshot [9], diffusion pseudotime [43] and veloVI [13] are applied on simulation data used in Fig 2. Inferred time is plotted against true time, where x axis is the true simulation time and y axis is corresponding inferred time normalized between 0 and 1. RMSE stands for root mean square error of inferred time.

(PDF)

**S24 Fig. Results of other methods on Neuron data**. Monocle 3 [6], Slingshot [9], diffusion pseudotime [43] and veloVI [13] are applied on Neuron data used in S21 Fig to generate violin plots comparing inferred time to experimental time. In these plots, the x-axis represents the experimental time, while the y-axis shows the corresponding inferred time.

(PDF)

## Acknowledgments

We thank Maria Carilli, Tara Chari, and Catherine Felce for helpful discussions and feedback on the manuscript. We thank Kaushik Roy for careful reading of the manuscript and helpful comments.

## Author contributions

**Conceptualization:** Gennady Gorin, Lior Pachter

**Formal analysis:** Meichen Fang, Gennady Gorin, Lior Pachter

**Funding acquisition:** Lior Pachter

**Investigation:** Meichen Fang, Gennady Gorin, Lior Pachter

**Methodology:** Meichen Fang, Gennady Gorin, Lior Pachter

**Software:** Meichen Fang

**Supervision:** Lior Pachter

**Writing – original draft:** Meichen Fang, Lior Pachter

**Writing – review & editing:** Meichen Fang, Gennady Gorin, Lior Pachter

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
