## [Decision Letter · Decision Letter 0]

14 Aug 2024

Dear Ms Fang,

Thank you very much for submitting your manuscript "Trajectory inference from single-cell genomics data with a process time model" for consideration at PLOS Computational Biology.

As with all papers reviewed by the journal, your manuscript was reviewed by members of the editorial board and by several independent reviewers. In light of the reviews (below this email), we would like to invite the resubmission of a revised version that takes into account the reviewers' comments.

In particular, we emphasize the importance of clarifying the assumption of a known trajectory structure. We recommend a demonstration and/or discussion of robustness where an inappropriate trajectory structure is specified. 

We cannot make any decision about publication until we have seen the revised manuscript and your response to the reviewers' comments. Your revised manuscript is also likely to be sent to reviewers for further evaluation.

Sincerely,

Jean Fan

Academic Editor

PLOS Computational Biology

Jian Ma

Section Editor

PLOS Computational Biology

Reviewer's Responses to Questions

**Comments to the Authors:**

Reviewer #1: This manuscript presents the Chronocell method for single-cell RNA-seq data trajectory inference and proposes a "process time" concept, which provides more physical meaning compared to the widely acknowledged "pseudotime" concept. To enhance the manuscript's clarity and impact, I recommend addressing the following points:

(1) Clarify the number of topologies that the proposed method can infer.

(2) Provide a detailed, step-by-step explanation of the process from the gene expression matrix to the derivation of trajectory and process time.

(3) Test the method on more complex trajectories and with a larger number of cells. The trajectory illustrated in Figure 2, which only shows one bifurcation, is relatively simple.

(4) Evaluate the computational efficiency of the iterative procedure for datasets at the million-cell level. Assess the running time as the number of cells increases significantly, given the advancements in single-cell sequencing technology.

(5) Include a comprehensive comparison of the proposed method with recent state-of-the-art methods to further demonstrate its effectiveness.

(6) Expand the discussion comparing "process time" and "pseudotime," which is currently limited to Figure 2. While both trajectories for Slingshot and the proposed method appear correct, a more detailed comparison is essential to highlight the advantages of the proposed "process time."

Reviewer #2: This paper proposes a trajectory inference method, Chronocell, incorporating RNA velocity and trajectory inference. The method aims to achieve the following few goals: 1. provide a unified model for RNA velocity and trajectory inference to improve intepretability of the model and results. 2. Separate between cell population that has a continuous trajectory structure from cell population that has discrete cluster-like structure. 3. Provide estimation of transcription related parameters such as degradation rates. I think the model proposed in the paper is interesting, though I have some concerns in the assumptions of the model. I also think the authors need more benchmarking results comparing with existing trajectory inference and RNA velocity methods to show benefits of the new approach.

Here are my specific comments:

1. The method assumes a known trajectory structure (a directed tree with known number of nodes, Page 25 line 525). I think this can be a big assumption but I'm not sure how much the authors need to assume from reading the methodology section. For example, does the method require knowing the shape of the directed tree (the number of branches at each stage) or just the number of nodes? Typically, even if there are biological prior knowledge, you would not know the exact shape of the trajectory structure, especially for complicated developmental processes. Most of the trajectory inference methods do not need to assume that. I think the authors need to clarify this assumption.

Additionally, many trajectory structures are not directed trees. For instance, it is common that there are diconnected cell states/types from the main branch (Figure 1c, Sealens et. al. Nature Biotech 2019). In Sec 2.5, the authors have applied Chronocell on a cell population with a cell cycle structure which is not a directed tree. I'm confused on how Choronocell can be applied to that scenario.

2. A question related to the above one. The authors claim that they can separate between cell population that has a continuous trajectory structure from cell population that has discrete cluster-like structure. How about cell population that has a subset of cells coming from a continuous trajectory and a subset of cells that come from discrete clusters? This scenario would happen often in practice.

3. The authors assume constant transcription rate for each gene within a lineage and different transcription rate for each lineage. Are there any biological or empirical support on that assumption? Earlier empirical results seems to have shown that assumptions on the transcription rate can greatly affect the performance of RNA velocity method, can the authors provide further analysis on whether Chronocell is sensitive to the assumption on transcription rate?

The authors have also assumed that all selected genes have completely synchronized switch time of transcription rate. Can the authors provide some justification on this?

4. As the authors discussed, some datasets have cells that are collected at a sequence of ordered time points. Can the method incorporate such information into the analysis?

5. Since there are many trajectory inference and RNA velocity methods and highly cited benchmarking paper on trajectory inference, I think the authors need to perform more benchmarking analysis to show the benefits of their method. For example, in the simulations, the authors can also generate synthetic data from existing software such as dyngen. In the real data analysis, the authors may compare with widely used methods such as slingshot and Monocle, existing methods combing trajectory inference and RNA velocity, and consider datasets that have more complicated trajectory structures. Currently, I have not seen a very clear evidence on why the users should use Choronocell instead of existing tools.

6. How does the method select DE genes along the estimated trajectory structure? Will the method avoid selecting false positive genes due to the fact that the process time (pseudotime) is estimated?

7. A minor technical question: the notation of the model in section 4.1.2 is confusing. j stands for a gene, so \lambda_c should have an index of j instead of c? The method seems to assume that each gene as a gene specific transcription rate, degradation rate a splicing rate?

8. Personally, I feel that some of the figures are too technical and might be hard to follow even if one has read the text multiple times. For example, I'm not sure if I understand the Chronocell part in Figure 2. Are the figures just illustrations? I can not understand the meaning of the numbers in each figure and there are no explanations. In figure 4, do the authors use Figure 4b to pick the best initialization? which one did they pick? Why does Figure 4 show that method performs very well and people should use it?

**Have the authors made all data and (if applicable) computational code underlying the findings in their manuscript fully available?**

Reviewer #1: Yes

Reviewer #2: Yes

PLOS authors have the option to publish the peer review history of their article (what does this mean?). If published, this will include your full peer review and any attached files.

Reviewer #1: No

Reviewer #2: No
---

## [Decision Letter · Decision Letter 1]

25 Dec 2024

Dear Ms Fang,

We are pleased to inform you that your manuscript 'Trajectory inference from single-cell genomics data with a process time model' has been provisionally accepted for publication in PLOS Computational Biology.

Best regards,

Jean Fan

Academic Editor

PLOS Computational Biology

Jian Ma

Section Editor

PLOS Computational Biology

Reviewer's Responses to Questions

**Comments to the Authors:**

Reviewer #1: The authors have addressed most of my concerns. However, I still have the following concern:

(1) The proposed Chronocell method assumes that the user have prior knowledge of the trajectory topology. However, I believe the primary goal of applying a trajectory inference method is to uncover both the topology and the pseudotime of the cells. Furthermore, the comparison with baseline methods appears unfair in this context, given that Slingshot, Monocle 3, and Scanpy do not rely on the ground truth topology—an essential piece of information for the trajectory inference task.

Reviewer #2: I'm OK with the revision provided by the authors. I think they have addressed all my concerns.

**Have the authors made all data and (if applicable) computational code underlying the findings in their manuscript fully available?**

Reviewer #1: Yes

Reviewer #2: None

PLOS authors have the option to publish the peer review history of their article (what does this mean?). If published, this will include your full peer review and any attached files.

Reviewer #1: No

Reviewer #2: No

---

## [Editor Report · Acceptance letter]

PCOMPBIOL-D-24-01026R1

Trajectory inference from single-cell genomics data with a process time model

Dear Dr Fang,

I am pleased to inform you that your manuscript has been formally accepted for publication in PLOS Computational Biology. Your manuscript is now with our production department and you will be notified of the publication date in due course.

With kind regards,

Zsofia Freund
